# SIREN: Subgraph Isomorphism via Reinforcement Enhanced Graph Neural Networks

## Abstract

The subgraph isomorphism problem comprises two distinct objectives: (a) Existence determination: Verifying whether an input graph contains a subgraph isomorphic to another input graph; and (b) Complete solution enumeration: Outputting the exhaustive set of all isomorphic mappings when they exist. Solving this problem serves as a fundamental requirement for numerous application domains. However, as an NP-complete problem, existing mainstream solvers primarily rely on heuristic techniques, demonstrating limited efficiency when handling large-scale input graphs. To address this challenge, we propose SIREN - a graph neural network enhanced with deep reinforcement learning for subgraph isomorphism resolution. SIREN establishes graph embeddings through partial order-aware GNNs, while employing Deep Q-Networks with bidomain-based pruning to accelerate the graph matching process. Experimental results on real-world datasets demonstrate that SIREN achieves 100% precision with modest computational time, outperforming AI-based approximate matching methods. Compared to state-of-the-art exact solvers, SIREN delivers $\sim 52\%$ faster execution than leading AI approaches and $\sim 21\%$ acceleration over top heuristic methods.

## 1 Introduction

The representation of structured data using graphs has evolved over decades as a foundational methodology in data modeling West et al. (2001). Meanwhile, graph-based algorithms have found widespread adoption for analyzing complex relational patterns across scientific and industrial domains Dijkstra (2022); Kruskal (1956). Within graph theory, the subgraph isomorphism problem represents a core computational challenge Nguyen et al. (2022); Zhang et al. (2024). Specifically, it requires determining whether a given graph contains a structurally isomorphic subgraph of another graph, while also deriving the explicit node correspondence when such isomorphism exists.

The subgraph isomorphism problem has garnered significant attention across pivotal domains due to its critical role in enabling structural pattern analysis. In biomolecular sciences, it underpins the determination of structural compatibility between molecules and proteins for drug discovery and protein interaction studies Balaban (1985); Bonnici et al. (2013). Within semantic web technologies, it facilitates efficient Resource Description Framework (RDF) query processing to traverse complex knowledge graphs Kim et al. (2015). Social network analytics leverages subgraph isomorphism detection to generate personalized recommendation systems through dynamic community subpattern mining Rong et al. (2018). Furthermore, in domain-specific computing architectures, this capability proves essential for optimizing loop mapping schemes in reconfigurable computing systems, where topological constraints demand rigorous subgraph matching Hamzeh et al. (2012). These cross-disciplinary applications collectively demonstrate the problem's fundamental importance in modern computational paradigms.

The subgraph isomorphism problem, known to be NP-complete Conte et al. (2004), is classically tackled using heuristic methods, which can be broadly classified into three categories: (1) **Tree search algorithms**, ranging from classical ones (Ullmann Ullmann (1976), VF-series Carletti et al. (2017)) to modern extensions (RM Sun et al. (2020), VEQ Kim et al. (2021), CaLiG Yang et al. (2023)), that employ depth-first search combined with pruning techniques; (2) **Constraint programming frameworks**, which model the problem as a constraint satisfaction problem (CSP) using integer linear programming (ILP), SAT, or other formalisms (e.g., McGregor McGregor (1979), Solnon

Solnon (2010), Zampelli Zampelli et al. (2010)); (3) **Graph indexing approaches**, often inspired by database systems (e.g., GraphQL He & Singh (2008), QuickSI Shang et al. (2008), GADDI Zhang et al. (2009)), that use precomputed structural signatures and inverted indices to accelerate filtering. Despite the use of advanced pruning strategies, state-of-the-art heuristic solvers still suffer from exponential worst-case time complexity, which often limits their practicality on large real-world graphs.

Recent advances in artificial intelligence, particularly graph neural networks (GNNs) designed for graph isomorphism analysis Xu et al. (2019), have spurred new learning-based approaches for subgraph isomorphism. Representative methods include IsoNet Roy et al. (2022), streaming models Duong et al. (2021), GMN Li et al. (2019), GNN-PE Ye et al. (2024b), EinsMatch Ramachandran et al. (2024), and SubMDSE Raj et al. (2025). Furthermore, AI techniques have been extended to address related problems such as subgraph alignment Bainson et al. (2024), graph edit distance computation Piao et al. (2023), and maximum common subgraph identification Bai et al. (2021).

However, current learning-based methods exhibit notable limitations: they often yield probabilistic approximations rather than exact solutions Ramachandran et al. (2024), or are confined to very small query graphs (e.g., $\leq 10$ nodes) Ye et al. (2024b). A more fundamental constraint is their heavy reliance on solver-generated labels produced by traditional algorithms for supervised training, which introduces computational bottlenecks and restricts practical deployment.

To address the limitations of existing approaches, we present **SIREN** (Subgraph Isomorphism via Reinforcement-Enhanced Graph Neural Networks). Our framework integrates a Deep Q-Network (DQN) Volodymyr et al. (2019) with bidomain-based pruning to autonomously discover optimal node selection heuristics, which are critical components in state-of-the-art subgraph isomorphism solvers. Complementing this, SIREN employs a pretrained graph neural network grounded in partial order relation learning to hierarchically encode subgraph structural dependencies. While maintaining probabilistic completeness, experimental results demonstrate that SIREN outperforms all machine learning-based methods in accuracy for approximate matching tasks. Simultaneously, it achieves significant efficiency gains over both heuristic approaches and AI-based methods when enumerating complete solution sets. The primary contributions of this paper are as follows:

1. **DQN-GNN integration with probabilistic completeness:** We propose a novel DQN-GNN integration method with bidomain-based pruning that efficiently solves subgraph isomorphism problems while guaranteeing probabilistic completeness.

2. **Partial-order-aware pretrained GNN:** We introduce a partial-order-aware GNN pretraining strategy that enhances substructure relationship understanding.

3. **Unified framework:** Our framework simultaneously addresses both existence determination and complete solution enumeration for subgraph isomorphism.

4. **Superior precision and efficiency:** Experimental results on real-world datasets demonstrate that SIREN achieves $100\%$ precision with modest computational time, outperforming AI-based approximate matching methods. Compared to state-of-the-art exact solvers, SIREN delivers $\sim 52\%$ faster execution than leading AI approaches and $\sim 21\%$ acceleration over top heuristic methods.

## 2 PRELIMINARIES

**1) Graph:** A graph can be formally defined as $G = (V, E)$, where $V = \{v_1, \ldots, v_n\}$ denotes the finite set of vertices and $E \subseteq V \times V$ denotes the edge set. Each edge $e_{ij} \in E$ connects two vertices $v_i, v_j$ in $V$.

**2) Graph Isomorphism:** Two graphs $G_1 = (V_1, E_1)$ and $G_2 = (V_2, E_2)$ are isomorphic ($G_1 \cong G_2$) if there exists a bijective mapping $f : V_1 \xrightarrow{\sim} V_2$ such that:

$$\forall v_i, v_j \in V_1, \quad (v_i, v_j) \in E_1 \iff (f(v_i), f(v_j)) \in E_2 \tag{1}$$

**3) Subgraph:** Let $G = (V, E)$ be a graph. A subgraph $G' = (V', E')$ of $G$ is defined as $G' \sqsubseteq G$, which satisfies:

$$V' \subseteq V \quad \text{and} \quad E' \subseteq \big\{(u, v) \in E \mid u, v \in V'\big\} \tag{2}$$

Figure 1: Overview of the framework of SIREN. (a) NN Training Framework, (b) Reinforcement Learning-Driven Search Framework, (c) Partial Order-Aware GNN-based DQN Architecture

**4) Subgraph Isomorphism Problem:** The subgraph isomorphism problem involves determining whether there exists a subgraph $G' \sqsubseteq G_T$ such that $G_Q \cong G'$, where $G_Q$ denotes the query graph and $G_T$ denotes the target graph. Formally:

$$\exists\, G' = (V', E') \sqsubseteq G_T \quad \text{s.t.} \quad G_Q \cong G' \tag{3}$$

If such subgraphs exist, the solution set $\mathcal{S}_G$ comprises all valid subgraphs:

$$\mathcal{S}_G = \left\{ G' \sqsubseteq G_T \mid G_Q \cong G' \right\} \tag{4}$$

## 3 THE SIREN METHOD

In this section, we provide a detailed description of the SIREN framework, which stands for Subgraph Isomorphism via Reinforcement-Enhanced Graph Neural Networks. Section 3.1 presents our DQN-based probabilistic complete framework for subgraph isomorphism, and Section 3.2 introduces our partial order relation-aware GNN architecture.

### 3.1 PROBABILISTIC COMPLETE FRAMEWORK FOR SUBGRAPH ISOMORPHISM

SIREN addresses the subgraph isomorphism problem by integrating graph neural networks with reinforcement learning. This approach formulates the problem as a Markov Decision Process (MDP) Puterman (1990) and employs a DQN-based framework Volodymyr et al. (2019) to enhance traditional tree-search heuristics Carletti et al. (2017). Our search framework guarantees the completeness of SIREN, with a detailed proof provided in Section A.4.

**1) DQN-Enhanced Search Framework.** The DQN-enhanced search framework of SIREN, depicted in Figure 1(b), maintains a state heap $\mathcal{ST}$ to store feasible states. At each decision step, the state $s_t$ with the maximal action space cardinality $|\mathcal{A}_t|$ is selected as the current state, prioritizing branches with higher combinatorial potential. At each step, the agent either adds a new node pair to the current partial matching or backtracks from a previous decision. The selection of candidate nodes is optimized using a partial order-aware Graph Isomorphism Network (GIN) integrated within the DQN. The search process terminates when all possible isomorphisms are found or non-existence is proven. SIREN can also be configured to terminate upon discovering the first feasible solution, which is suitable for applications requiring only a single valid subgraph matching.

**2) DQN Framework.** As illustrated in Figure 1(c), the framework utilizes continuous embedding representations to encode states $s_t$ and actions $a_t$. These representations are processed by a DQN architecture that consists of a partial-order aware GNN encoder and learnable projection modules. The DQN maps state-action pairs $(s_t, a_t)$ to Q-value estimates $Q(s_t, a_t)$, thereby enabling data-driven policy optimization.

**3) State Representation.** A state $s_t$ comprises the node-node mapping $\mathcal{M}_t$ between the selected subgraphs, the input graphs themselves, and the bidomain information corresponding to the current mapping $\mathcal{M}_t$. The features provided to the GNN model for state $s_t$ include node types/labels, (k-hop) local degree profiles, matching status indicators, and edge information of the graphs.

**4) Action Space.** At each step, the agent selects (1) a node pair $(v_q, v_t) \in \mathcal{C}_t$ that maintains topological consistency with $\mathcal{M}_t$, and (2) a special $\langle \text{terminate} \rangle$ action to prune unpromising branches,

where $\mathcal{C}_t$ denotes the candidate node pairs for expansion. The action space size is dynamic: $|\mathcal{C}_t| + 1$ (including termination).

**5) Action Embedding.** SIREN incorporates an action embedding optimization mechanism. For each feasible node pair $(v_q, v_t)$, the associated node features are projected into a low-dimensional embedding space via a learnable matrix $\mathbf{W}_a$. This embedding captures the essential value characteristics of the action, which in turn refines the Q-value computation.

**6) Reward Function.** The reward function of SIREN is designed to guide the learning process through a dense reward mechanism:

- A base reward of $+\alpha \cdot 1/|V_Q|$ for each valid node pair matching (default $\alpha = 0.5$), where $|V_Q|$ is the number of nodes in the query graph.
- An incremental structural reward of $+\beta \cdot \Delta E/|E_T|$ for each newly matched edge (default $\beta = 0.5$), where $\Delta E$ denotes the increase in the number of correctly matched edges and $|E_T|$ is the total number of edges in the target graph.
- A sparse success reward of $+\gamma \cdot |V_Q|$ (default $\gamma = 0.05$) upon finding a complete isomorphism.

**7) Bidomain-Based Embeddings and Pruning.** SIREN leverages Bidomains Bai et al. (2021) to enhance its search process in two key ways: by providing additional graph embeddings that inform the Q-value computation, and by facilitating efficient pruning through estimates of the exploration upper bound. For state $s_t = \mathcal{M}_t = (\mathcal{M}_Q, \mathcal{M}_T)$ with matched sets $\mathcal{M}_Q \subseteq V(G_Q)$, $\mathcal{M}_T \subseteq V(G_T)$, the $k$-th bidomain $\mathcal{B}_k$ is defined as:

$$\mathcal{B}_k = \langle \mathcal{V}_k^{\mathrm{Q}}, \mathcal{V}_k^{\mathrm{T}} \rangle \tag{5}$$

where $\mathcal{V}_k^{\mathrm{Q}} \subseteq V(G_Q)$ and $\mathcal{V}_k^{\mathrm{T}} \subseteq V(G_T)$ satisfy:

$$\forall u \in \mathcal{V}_k^{\mathrm{Q}}, v \in \mathcal{V}_k^{\mathrm{T}} : \mathrm{adj}(u, \mathcal{M}_Q) \equiv \mathrm{adj}(v, \mathcal{M}_T) \tag{6}$$

denoting identical connectivity patterns to matched sets $\mathcal{M}_Q \subseteq V(G_Q)$ and $\mathcal{M}_T \subseteq V(G_T)$. A detailed description of the bidomain technique is provided in Section A.8.

**8) Embedding Fusion.** We perform embedding fusion on the embeddings $\mathbf{h}_{G_Q}$ and $\mathbf{h}_{G_T}$ generated by the GNN for the query graph and target graph, respectively. This alignment operation involves processing each embedding through a 1D convolutional layer followed by a pooling layer, and subsequently merging the resulting representations.

**9) Action-Value Function.** For state transition, candidate actions $\mathcal{A}_t = \{(u, v) \in \mathcal{C}_t\}$ are evaluated by the DQN's action value function $Q_\theta(s_t, a_t)$, where $\mathbf{h}$ denotes the graph embeddings generated with our trained GNN model:

$$Q_\theta(s_t, a_t) = \mathcal{F}\Big( \underbrace{\mathrm{GNN}_{\mathrm{enc}}(G_Q, G_T, s_t, \mathcal{B}_{s_t})}_{\text{state embedding}} \oplus \underbrace{\mathbf{W}_a \psi(a_t)}_{\text{action embedding}} \Big)$$

$$= \mathcal{F}(\mathbf{h}_{s_t}, \mathrm{Fuse}(\mathbf{h}_{G_Q}, \mathbf{h}_{G_T}), \mathbf{h}_{BD}, a_t) \tag{7}$$

where $\psi(a_t)$ is the feature vector of action $a_t$, and $\mathbf{W}_a$ denotes learnable action projection weights.

**10) $\epsilon$-Greedy Action Selection.** The agent selects an action $a_t$ at each timestep using an $\epsilon$-greedy policy based on the current Q-value estimates:

$$a_t^* = \begin{cases} \arg\max_{a_t \in \mathcal{A}_t}(Q_\theta(s_t, a_t)) & \text{with probability } 1 - \epsilon \\ \text{random action} & \text{with probability } \epsilon \end{cases}$$

The updated state $s_{t+1} = \mathcal{M}_t \cup a_t^*$ is is then pushed back onto $\mathcal{ST}$.

**11) Action Space Partition.** For large-scale graphs where the number of candidate matching actions (node pairs) becomes prohibitively large, we partition the action space into chunks and compute Q-values separately for each chunk. This approach prevents GPU memory overflow while maintaining computational efficiency.

The training process of our DQN model consists of three consecutive phases: pretraining, imitation learning, and reinforcement learning, which is detailed in Section A.5. To analyze the impact of different reinforcement learning paradigms, we conducted comparisons between DQN and Proximal Policy Optimization (PPO) Schulman et al. (2017), as detailed in Section A.11.

## 3.2 Partial Order-Aware GNN Architecture

The subgraph relationship satisfies the properties of reflexivity, transitivity, and antisymmetry, thereby establishing it as a partial order on the set of graphs. Formally:

- **Reflexivity:** $G \sqsubseteq G, G \cong G$.
- **Transitivity:** $(G_1 \sqsubseteq G_2) \wedge (G_2 \sqsubseteq G_3) \implies G_1 \sqsubseteq G_3$.
- **Antisymmetry:** $(G_1 \sqsubseteq G_2) \wedge (G_2 \sqsubseteq G_1) \iff G_1 = G_2$.

The formal proof of these properties is provided in Section A.1. Moreover, The intersection of the set of $G_1$'s subgraphs and the set of $G_2$'s subgraphs contains all common subgraphs of $G_1$ and $G_2$ Lou et al. (2020). Therefore, SIREN employs a partial order-aware GNN to enforce that the learned graph embeddings preserve the isomorphic partial order relationships within the embedding space. This geometric constraint ensures that for any two graphs $G_Q$ and $G_T$ with embeddings $\mathbf{h}_{G_Q} = (\mathbf{h}_{G_Q}^1, ..., \mathbf{h}_{G_Q}^D)$ and $\mathbf{h}_{G_T} = (\mathbf{h}_{G_T}^1, ..., \mathbf{h}_{G_T}^D)$:

$$\forall i \in 1, ..., D, \mathbf{h}_{G_Q}^i \leq \mathbf{h}_{G_T}^i \iff G_1 \sqsubseteq G_2 \tag{8}$$

The proof that our GNN model satisfies the partial order relationships described in Equation 8 is provided in Section A.3. To preserve the partial order relationship of the subgraphs, we use the max margin loss to train our GNN model. Within each minibatch, we define $P = \{(q, t) \in \mathcal{MB} \mid \mathcal{N}(q) \sqsubseteq \mathcal{N}(t)\}$ as the set of positive pairs where the neighborhood subgraph of query node $q$ is isomorphic to a subgraph of target node $t$'s neighborhood, and $N = \mathcal{MB} \setminus P$ as the negative pairs violating this structural constraint. The loss function $\mathcal{L}$ then operates on these sets to enforce geometric consistency in the embedding space:

$$\mathcal{L}(\mathbf{h}_q, \mathbf{h}_t) = \sum_{\mathbf{h}_q, \mathbf{h}_t \in P} E(\mathbf{h}_q, \mathbf{h}_t) + \sum_{\mathbf{h}_q, \mathbf{h}_t \in N} \max(0, \alpha - E(\mathbf{h}_q, \mathbf{h}_t)) \tag{9}$$

where

$$E(\mathbf{h}_q, \mathbf{h}_t) = || \max(0, \mathbf{h}_q - \mathbf{h}_t) ||_2^2 \tag{10}$$

We employed an improved GIN model Xu et al. (2019), incorporating multi-scale feature fusion techniques to generate graph embedding vectors. For layer $l \in \{0, 1, ..., L-1\}$, the embedding of node $v$ is computed as:

$$\mathbf{h}_v^{(l+1)} = \text{MLP}^{(l)}((1 + \epsilon^{(l)}) \cdot \mathbf{h}_v^{(l)} + \sum_{u \in \mathcal{N}(v)} \mathbf{h}_u^{(l)}) \tag{11}$$

where $\epsilon^{(l)} \in \mathbb{R}$ is a learnable scalar parameter, $\mathcal{N}(v)$ denotes the neighborhood of node $v$, and $\text{MLP}^{(l)}$ denotes the multi-layer perceptron with LeakyReLU Maas et al. (2013) activation $\sigma$. Let $\gamma_l$ be the learnable hierarchical weight coefficient, the graph embedding is obtained by concatenating sum-pooled features across layers:

$$h_G = \sum_{l=0}^{L} \gamma_l \cdot \sum_{v \in V} h_v^{(l)} \tag{12}$$

The training protocol of our partial order-aware GNN model is detailed in Section A.6.

## 4 Experiments

To evaluate the effectiveness and efficiency of SIREN, we compared SIREN with 19 state-of-the-art neural network-based methods and heuristic methods. The experiments are conducted real large graph datasets from TUDataset Morris et al. (2020) and on synthetic graph datasets.

### 4.1 Baseline Methods

We compared SIREN with 10 state-of-the-art neural network-based methods for subgraph isomorphism, including SimGNN Bai et al. (2019), GraphSim Bai et al. (2020), GEDGNN Piao et al. (2023), GOTSim Doan et al. (2021), ERIC Zhuo & Tan (2022), NeuroMatch Lou et al. (2020), GMN Li et al.

Table 1: Evaluation of Mean Average Precision (MAP) on real-world graph pairs. $K$ = layers, $|ND|$ = node state dimensions. Each dataset contains 300 query graphs and 800 target graphs.

| Method | $K$ | $|ND|$ | AIDS | MUTAG | PTC-FM | PTC-FR | PTC-MM | PTC-MR |
|---|---|---|---|---|---|---|---|---|
| Average $|V_Q|$ | | | 11.61 | 12.91 | 11.73 | 11.81 | 11.80 | 11.87 |
| Max $|V_Q|$ | | | 15 | 15 | 15 | 15 | 15 | 15 |
| Average $|V_T|$ | | | 18.50 | 18.41 | 18.30 | 18.32 | 18.36 | 18.32 |
| Max $|V_T|$ | | | 20 | 20 | 20 | 20 | 20 | 20 |
| SIMGNN | 3 | 10 | $0.326 \pm 0.019$ | $0.303 \pm 0.012$ | $0.416 \pm 0.015$ | $0.355 \pm 0.015$ | $0.358 \pm 0.015$ | $0.308 \pm 0.017$ |
| GRAPHSIM | 3 | 10 | $0.173 \pm 0.007$ | $0.182 \pm 0.008$ | $0.231 \pm 0.011$ | $0.165 \pm 0.007$ | $0.2 \pm 0.009$ | $0.216 \pm 0.013$ |
| GEDGNN | 3 | 10 | $0.340 \pm 0.015$ | $0.605 \pm 0.029$ | $0.437 \pm 0.013$ | $0.497 \pm 0.018$ | $0.509 \pm 0.018$ | $0.309 \pm 0.009$ |
| GOTSIM | 5 | 10 | $0.336 \pm 0.017$ | $0.387 \pm 0.018$ | $0.459 \pm 0.017$ | $0.361 \pm 0.013$ | $0.417 \pm 0.017$ | $0.430 \pm 0.017$ |
| ERIC | 5 | 10 | $0.512 \pm 0.022$ | $0.558 \pm 0.027$ | $0.624 \pm 0.019$ | $0.572 \pm 0.021$ | $0.573 \pm 0.02$ | $0.639 \pm 0.018$ |
| GMN-MATCH | 3 | 10 | $0.609 \pm 0.02$ | $0.693 \pm 0.026$ | $0.686 \pm 0.018$ | $0.667 \pm 0.021$ | $0.627 \pm 0.02$ | $0.683 \pm 0.017$ |
| NEUROMATCH | 3 | 10 | $0.454 \pm 0.025$ | $0.583 \pm 0.027$ | $0.622 \pm 0.019$ | $0.572 \pm 0.023$ | $0.522 \pm 0.019$ | $0.565 \pm 0.02$ |
| ISONET | 3 | 10 | $0.704 \pm 0.021$ | $0.733 \pm 0.023$ | $0.782 \pm 0.017$ | $0.734 \pm 0.02$ | $0.758 \pm 0.016$ | $0.764 \pm 0.015$ |
| SUBMDSE-LATE | 5 | 10 | $0.712 \pm 0.018$ | $0.721 \pm 0.025$ | $0.793 \pm 0.016$ | $0.744 \pm 0.019$ | $0.758 \pm 0.015$ | $0.782 \pm 0.014$ |
| SUBMDSE-EARLY | 5 | 10 | $0.817 \pm 0.017$ | $0.837 \pm 0.02$ | $0.887 \pm 0.012$ | $0.854 \pm 0.013$ | $0.849 \pm 0.012$ | $0.864 \pm 0.011$ |
| SIREN-MINI | 3 | 10 | $\mathbf{1.000 \pm 0.000}$ | $\mathbf{1.000 \pm 0.000}$ | $\mathbf{1.000 \pm 0.000}$ | $\mathbf{1.000 \pm 0.000}$ | $\mathbf{1.000 \pm 0.000}$ | $\mathbf{1.000 \pm 0.000}$ |
| Avg. Runtime (ms) | | | 41.3 | 33.4 | 35.5 | 43.7 | 34.8 | 44.2 |
| SIREN | 8 | 64 | $\mathbf{1.000 \pm 0.000}$ | $\mathbf{1.000 \pm 0.000}$ | $\mathbf{1.000 \pm 0.000}$ | $\mathbf{1.000 \pm 0.000}$ | $\mathbf{1.000 \pm 0.000}$ | $\mathbf{1.000 \pm 0.000}$ |
| Avg. Runtime (ms) | | | 50.5 | 47.3 | 42.5 | 56.3 | 52.4 | 52.7 |
| Improvement | | | 18.3% | 16.3% | 11.3% | 14.6% | 15.1% | 13.6% |

(2019), IsoNet Roy et al. (2022), SubMDSE Raj et al. (2025) and GNN-PE Ye et al. (2024b), as well as 9 heuristic state-of-the-art methods for solving the subgraph isomorphism problem, including RI Bonnici et al. (2013), VF2++ Jüttner & Madarasi (2018), GraphQL He & Singh (2008), QuickSI Shang et al. (2008), VF3 Carletti et al. (2017), RM Sun et al. (2020), VEQ Kim et al. (2021), CaLiG Yang et al. (2023), and DPIso Han et al. (2019).

Existing neural network-based subgraph isomorphism methods can be divided into two categories: exact matching and approximate prediction. Exact matching methods return the precise solution set for the subgraph isomorphism problem, while approximate prediction methods provide a quick assessment of whether two graphs satisfy the subgraph isomorphism relationship. Among these methods, GNN-PE is an exact matching method, while the other 7 methods are approximate prediction methods. To ensure a fair comparison, we evaluated the solving speed against GNN-PE and heuristic methods (while the accuracy of all methods is inherently 1.000), and compared the solving accuracy with the other 7 methods, using the same real-world datasets and sampling settings as in the original papers.

## 4.2 EXPERIMENTAL SETTINGS

In SIREN, we utilize 8 layers of Graph Isomorphism Networks (GIN) Xu et al. (2019) each with 64 dimensions for the embeddings. For DQN, we use MLP layers to project concatenated embeddings to a scalar. The discount factor $\gamma$ of the DQN is set to 1.0, and the learning rate of the DQN and the GIN is set to 0.001. The models are trained using the Adam optimizer Kingma (2014). The learning rate is annealed with a cosine annealer with restarts every 100 epochs. The DQN is trained by 10000 iterations. Prior to DQN training, we conduct a 50000-epochs supervised pre-training of the GNN model to generate geometrically consistent partial order-preserving graph embeddings. The training data is generated by randomly sampling neighborhoods from large real-world graph datasets Morris et al. (2020), while being regenerated every 50 epochs. To ensure fairness across diverse tasks, we trained two distinct GNN configurations (in SIREN-Mini and SIREN). Detailed parameter specifications are provided in Table 1. The experiments were conducted on a Ubuntu server equipped with a 128-Core Intel(R) Xeon(R) Gold 5218 CPU running at 2.30 GHz and 256 GB of memory, along with $4\times$ Nvidia Ampere A800 GPU. SIREN were implemented with the PyTorch and PyTorch Geometric libraries Fey & Lenssen (2019).

## 4.3 EFFECTIVENESS OF SIREN

To compare SIREN with approximate prediction deep learning methods, we selected the same datasets as SubMDSE Raj et al. (2025) from TUDataset Morris et al. (2020) for testing, i.e., PTC-FR, PTC-FM, PTC-MM, PTC-MR, MUTAG, and AIDS. We compared SIREN with 9 state-of-the-art neural methods. The experimental results indicate that, without limiting the search time, SIREN achieved correct solutions for all datasets, with the longest time not exceeding 2 seconds ($\sim$1.8s). Compared

Table 2: Ablation Study of SIREN by Mean Average Precision (MAP) on Real-World Graph Pairs.

| Method | PTC-FR | PTC-FM | PTC-MM | PTC-MR | MUTAG | AIDS |
|---|---|---|---|---|---|---|
| SIREN-DQN | 0.880 | 0.893 | 0.961 | 0.828 | 0.952 | 0.904 |
| v.s. IsoNet | **+2.3%** | -0.8% | **+5.5%** | -6.6% | **+3.2%** | **+1.0%** |
| SIREN-GNN | 0.766 | 0.796 | 0.886 | 0.738 | 0.861 | 0.842 |
| v.s. IsoNet | -9.1% | -10.5% | -2.0% | -15.6% | -5.9% | -5.2% |
| SIREN | **1.000** | **1.000** | **1.000** | **1.000** | **1.000** | **1.000** |

Table 3: Runtime on Large-Scale Dense Real-World Graphs (Unit: s), OOM: Out of Memory.

| nodes of $G_T$ | 1000 | 2000 | 3000 | 4000 | 5000 | 10000 |
|---|---|---|---|---|---|---|
| nodes of $G_Q$ | 200 | 400 | 600 | 800 | 1000 | 2000 |
| Edge density | 0.4 | 0.3 | 0.2 | 0.2 | 0.2 | 0.2 |
| VF3 | 12151.6 | 16351.2 | 4292.6 | $> 10^5$ | $> 10^5$ | $> 10^5$ |
| VF3P | 1187.8 | 1923.6 | 636.2 | 24058.0 | 51020.4 | $> 10^5$ |
| VEQ | 860.9 | OOM | OOM | OOM | OOM | OOM |
| GNN-PE | $> 10^5$ | $> 10^5$ | $> 10^5$ | $> 10^5$ | $> 10^5$ | $> 10^5$ |
| GLSEARCH | 7534.5 | $> 10^5$ | 4682.1 | $> 10^5$ | $> 10^5$ | $> 10^5$ |
| SIREN | **642.4** | **943.2** | **486.7** | **16842.4** | **34719.5** | $> 10^5$ |

to the most accurate neural method SubMDSE-Early Raj et al. (2025), SIREN improved Mean Average Precision by an average of $14.9\%$. The experimental results show that SIREN outperforms approximate prediction deep learning methods in terms of effectiveness. These experimental results align with the theory, as SIREN guarantees that, given enough time, it can obtain an exact solution or prove that no valid solution exists. Furthermore, these results demonstrate that SIREN is a more effective method for general real-world datasets, as it provides acceptable running times and more accurate results.

## 4.4 EFFICIENCY AND THROUGHPUT RATE

To compare SIREN with exact matching deep learning methods and heuristic methods, we selected the same datasets as GNN-PE Roy et al. (2022) from TUDataset Morris et al. (2020) for testing, i.e., Yeast, Human, HPRD, WordNet, DBLP, Youtube, and US Patents. Statistics of these real graphs are summarized in Section A.10. We divided the dataset according to the method used in GNN-PE Roy et al. (2022), where the size of $|V(q)|$ is set to 8. Figure 2 shows the efficiency test results on large-scale datasets, which indicate that our SIREN method outperforms all existing heuristic methods across all test data, and it also surpasses GNN-PE in the majority of cases. Only for the DBLP dataset does SIREN ($\sim 0.19$s) perform slightly slower than GNN-PE ($\sim 0.17$s), with almost no difference. On average, compared to RI Bonnici et al. (2013), VF Jüttner & Madarasi (2018), GraphQL He & Singh (2008), QuickSI Shang et al. (2008), GNN-PE Roy et al. (2022), VF3 Carletti et al. (2017), RM Sun et al. (2020), VEQ Kim et al. (2021), CaLiG Yang et al. (2023), and DPIso Han et al. (2019), SIREN achieves speedups of $36.5\times$, $29.0\times$, $171.2\times$, $25.7\times$, $52.1\%$, $14.4\times$, $92.4\%$, $21.3\%$, $7.70\times$, and $55.3\%$, respectively.

Figure 3 compares the throughput, measured in generated matched embeddings per second (EPS), of SIREN against other methods on large-scale datasets. As shown, SIREN achieves a higher throughput than all baseline methods, outperforming the state-of-the-art approach VEQ by $1.046\times$.

## 4.5 SCALABILITY OF SIREN

To evaluate the scalability of SIREN, we compared SIREN with GNN-PE Roy et al. (2022), GLSearch Bai et al. (2021), VEQ Kim et al. (2021), and the state-of-the-art CPU-parallelized subgraph isomorphism method VF3p Carletti et al. (2017) on synthetic large adversarial dense graph dataset by VF3 Carletti et al. (2017). As shown in Table 3, experimental results on large graphs demonstrate that despite the server CPU's higher theoretical FP32 compute capacity (CPU: 140 TFLOPS vs. GPU: 40 TFLOPS), SIREN still outperforms traditional methods. On average, compared to

Figure 2: Efficiency of different methods.

Figure 3: Throughput Rate of different methods.

GNN-PE, GLSearch, VF3, VF3p and VEQ, SIREN achieves speedups of $8.32\times$, $4.82\times$, $3.34\times$, $47.0\%$, and $34.0\%$, respectively (excluding cases exceeding $10^5$ seconds or OOM). Our experiments further demonstrate that VEQ encounters out-of-memory failures under a 256GB memory constraint—already substantial for graph processing—when handling graphs exceeding 2000 nodes with edge density $\rho > 0.3$. This limitation stems from VEQ's significantly higher space complexity compared to SIREN, preventing it from scaling to large dense graphs.

### 4.6 ABLATION STUDY

Table 2 presents the ablation study results for two modules in SIREN: DQN and GNN. Here, SIREN-DQN refers to the results obtained using only the DQN + GNN structure, where GNN is randomly initialized without any pre-training method. SIREN-GNN indicates the scenario where the DQN structure is not used, and the NeuralMatch method Lou et al. (2020) is employed to directly output estimates for the subgraph isomorphism problem. The experimental results demonstrate that both the lack of GNN pre-training and the absence of the DQN model lead to a decline in solution quality.

## 5 RELATED WORK

### 5.1 TRADITIONAL APPROACHES

The subgraph isomorphism problem is known to be NP-complete and has traditionally been tackled using heuristic methods. These approaches can be broadly categorized as follows:

**1) Tree Search**: Classical algorithms such as Ullmann's algorithm Ullmann (1976) and the VF-series Cordella et al. (2001; 2004); Carletti et al. (2017) employ depth-first search combined with pruning heuristics (e.g., degree and label filters). VF3 Carletti et al. (2017) introduces state-space precomputation and look-ahead pruning to improve efficiency, particularly for large dense graphs. More recent methods, including RM Sun et al. (2020), VEQ Kim et al. (2021), and CaLiG Yang et al. (2023), further accelerate the resolution process.

**2) Constraint Programming**: These approaches model the problem as a Constraint Satisfaction Problem (CSP), using integer linear programming (ILP), Boolean satisfiability (SAT), or related formalisms. Variables typically represent query nodes, with domains consisting of candidate target nodes, and constraints encode structural requirements. Methods by McGregor McGregor (1979), Solnon Solnon (2010), and Zampelli Zampelli et al. (2010) employ arc consistency techniques to iteratively prune the solution space.

**3) Graph Indexing**: Methods inspired by database systems, such as GraphQL He & Singh (2008), QuickSI Shang et al. (2008), and GADDI Zhang et al. (2009), leverage precomputed structural signatures and inverted indices to enable efficient pre-match filtering and early termination.

**Limitations**: Despite their practical utility, all these methods exhibit exponential worst-case time complexity—typically $\mathcal{O}(n^k)$ for a query pattern of size $k$—and are sensitive to label noise and graph density. Indexing techniques also involve significant memory overhead, often reaching $\mathcal{O}(m^d)$ for depth-$d$ neighborhood features.

### 5.2 NEURAL APPROACHES FOR SUBGRAPH ISOMORPHISM

Recent neural network-based approaches for subgraph isomorphism can be divided into two main categories:

**1) Exact Solvers**: These methods aim to provably determine the existence of subgraph isomorphisms and recover corresponding node mappings. Representative techniques include subgraph index

embeddings Duong et al. (2021) and path dominance embeddings Ye et al. (2024b), which encode structural hierarchies to facilitate exact matching.

**2) Approximate Heuristics**: These methods prioritize scalability and efficiency at the cost of completeness, often relying on probabilistic or learned similarity measures. Examples include similarity-based networks Li et al. (2019), geometric embeddings Lou et al. (2020), D2Match Liu et al. (2023) and IsoNet Roy et al. (2022). More recent advances include EinsMatch Ramachandran et al. (2024), which introduces iterative alignment refinement, and SubMDSE Raj et al. (2025), which explores multifaceted design spaces for improved performance.

**Limitations**: Exact solvers typically depend on expensive solver-generated labels for supervised training, leading to high annotation and computational costs. On the other hand, approximate methods often lack theoretical guarantees and tend to exhibit reduced accuracy when applied to strict isomorphism problems.

### 5.3 NEURAL APPROACHES FOR RELATED PROBLEMS

**1) Graph Isomorphism**: While graph isomorphism can be solved in quasi-polynomial time, i.e., $O(e^{(\log n)^c})$ Babai (2016), it is often studied to characterize the expressive power of graph neural networks. Methods such as GIN Xu et al. (2019)—which simulates the Weisfeiler-Leman (WL) test Leman & Weisfeiler (1968)—and Graph Transformers Yun et al. (2019); Lee et al. (2024) have been developed, but they generally lack the precision required for subgraph isomorphism tasks.

**2) Maximum Common Subgraph (MCS)**: Subgraph isomorphism is a special case of MCS in which the common subgraph must be isomorphic to the query graph $G_Q$. Neural solvers such as GLSearch Bai et al. (2021) and MCSP+RL Liu et al. (2019) have been proposed for MCS, but they often fail to fully leverage the topological structure of the query graph, resulting in suboptimal efficiency.

**3) Subgraph Alignment**: This problem involves determining whether a query graph $G_Q$ is isomorphic to an induced subgraph of a target graph $G_T$. A recent spectral-based method Bainson et al. (2024) has been proposed to address it. Although subgraph alignment is a special case of subgraph isomorphism, the two are not identical; clarifying their distinctions is necessary to prevent confusion arising from terminological overlap.

**4) Graph Alignment**: Graph alignment aims to find a bijective mapping between the nodes of two graphs such that structural differences after mapping are minimized. FUGAL Bommakanti et al. (2024) introduces a learning-based approach for this task, while a differentiable top-kmethod Wang et al. (2023) addresses partial graph matching, a related relaxation.

**5) Graph Edit Distance (GED)**: Subgraph isomorphism can be reduced to GED by setting infinite substitution costs. However, general neural solvers for GED Raveaux (2021); Piao et al. (2023) are not well-suited for exact isomorphism due to their flexible cost models and broader objective.

**6) Large Language Model (LLM)-Based Methods**: Recent efforts such as ThinkOnGraph Sun et al. (2024) and GraphGPT Tang et al. (2024) focus primarily on attributed graphs. Although InstructGLM Ye et al. (2024a) encodes structural information through prompting, it—like other LLM-based approaches—has not yet shown effectiveness for exact combinatorial isomorphism problems.

## 6 CONCLUSION

The subgraph isomorphism problem is a challenging NP-complete problem with wide applications across various fields. In this paper, we introduce SIREN, an RL-enhanced GNN for subgraph isomorphism. Through our proposed DQN-based reinforcement learning framework and the GNN model based on partial order relations, we can improve the candidate node selection process in solving the subgraph isomorphism problem. Experiments on real datasets show that SIREN can effectively accelerate the solving of the subgraph isomorphism problem and enhance solution quality. Future work includes further improvements to the reinforcement learning framework and GNN, testing on more large-scale real datasets, and extending similar methods to other NP problems, such as the maximum clique problem.

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

## A  TECHNICAL APPENDICES AND SUPPLEMENTARY MATERIAL

### A.1  PROOFS OF PROPERTIES OF SUBGRAPH RELATIONSHIP

We provide proofs for the properties related to the subgraph relationship below, including reflexivity, transitivity, and antisymmetry.

**Theorem A.1** (Reflexivity). $G \sqsubseteq G$

*Proof.* Let $G = (V, E)$. Since $V \subseteq V$, $E \subseteq \{(u, v) \in E \mid u, v \in V\}$, it follows from the definition of the subgraph relationship that $G \sqsubseteq G$. □

**Theorem A.2** (Reflexivity of Graph Isomorphism). $G \cong G$

*Proof.* Consider a bijection $f : V \xrightarrow{\sim} V$ defined by $f(v) = v$ for all $v \in V$. This satisfies

$$\forall v_i, v_j \in V, \quad (v_i, v_j) \in E \iff (f(v_i), f(v_j)) = (v_i, v_j) \in E.$$

By the definition of graph isomorphism, $G$ and $G$ are isomorphic, i.e., $G \cong G$. □

**Theorem A.3** (Transitivity). $(G_1 \sqsubseteq G_2) \wedge (G_2 \sqsubseteq G_3) \implies G_1 \sqsubseteq G_3$.

*Proof.* Let $G_1 = (V_1, E_1)$, $G_2 = (V_2, E_2)$, and $G_3 = (V_3, E_3)$. From $G_1 \sqsubseteq G_2$, we have:

$$V_1 \subseteq V_2 \quad \text{and} \quad E_1 \subseteq \{(u, v) \in E_2 \mid u, v \in V_1\}.$$

Similarly, from $G_2 \sqsubseteq G_3$, we have:

$$V_2 \subseteq V_3 \quad \text{and} \quad E_2 \subseteq \{(u, v) \in E_3 \mid u, v \in V_2\}.$$

Since $V_1 \subseteq V_2$ and $V_2 \subseteq V_3$, it follows that:

$$V_1 \subseteq V_3.$$

For edges, from $E_1 \subseteq \{(u, v) \in E_2 \mid u, v \in V_1\}$ and $E_2 \subseteq \{(u, v) \in E_3 \mid u, v \in V_2\}$, we deduce:

$$\forall (u, v) \in E_1, \quad (u, v) \in E_2 \subseteq E_3 \quad \text{with} \quad u, v \in V_1 \subseteq V_3.$$

Therefore, we conclude:

$$E_1 \subseteq \{(u, v) \in E_3 \mid u, v \in V_1\}.$$

□

**Theorem A.4** (Antisymmetry). $(G_1 \sqsubseteq G_2) \wedge (G_2 \sqsubseteq G_1) \iff G_1 = G_2$.

*Proof.* Let $G_1 = (V_1, E_1)$ and $G_2 = (V_2, E_2)$. If $G_1 \sqsubseteq G_2$ and $G_2 \sqsubseteq G_1$, we have:

$$V_1 \subseteq V_2 \quad \text{and} \quad E_1 \subseteq \{(u, v) \in E_2 \mid u, v \in V_1\},$$

and

$$V_2 \subseteq V_1 \quad \text{and} \quad E_2 \subseteq \{(u, v) \in E_1 \mid u, v \in V_2\}.$$

From $V_1 \subseteq V_2$ and $V_2 \subseteq V_1$, we conclude $V_1 = V_2$. Similarly, $E_1 \subseteq E_2$ and $E_2 \subseteq E_1$ imply $E_1 = E_2$. Therefore, we have $G_1 = G_2$.

Conversely, if $G_1 = G_2$, then we have:

$$V_1 = V_2 \quad \text{and} \quad E_1 = \{(u, v) \in E_2 \mid u, v \in V_1\},$$

and

$$V_2 = V_1 \quad \text{and} \quad E_2 = \{(u, v) \in E_1 \mid u, v \in V_2\}.$$

Therefore, $G_1 \sqsubseteq G_2$ and $G_2 \sqsubseteq G_1$. □

## A.2 Corollaries of Subgraph Isomorphism Relationship

**Corollary A.5** (Subgraph Isomorphism Partial Order). *Building upon the properties mentioned in Section 3.2, the subgraph isomorphism relation $\preceq$:*

$$G_1 \preceq G_2 \iff \exists G' = (V', E') \sqsubseteq G_2 \quad s.t. \quad G_1 \cong G' \tag{13}$$

*satisfies the fundamental characteristics of a partial ordering on the graph space $\mathcal{G}$:*

- *Reflexivity:*

$$\forall G \in \mathcal{G}, \ G \preceq G$$

- *Transitivity:*

$$(G_1 \preceq G_2) \wedge (G_2 \preceq G_3) \implies G_1 \preceq G_3$$

- *Antisymmetry:*

$$(G_1 \preceq G_2) \wedge (G_2 \preceq G_1) \iff G_1 \cong G_2$$

We provide proofs for the properties related to the subgraph isomorphism below, including reflexivity, transitivity, and antisymmetry.

**Theorem A.6** (Reflexivity). $\forall G \in \mathcal{G}, \ G \preceq G$

*Proof.* By Theorem A.1, we have $G \sqsubseteq G$; by Theorem A.2, we have $G \cong G$, then we have $G \preceq G$. $\qquad\square$

**Theorem A.7** (Transitivity). $(G_1 \preceq G_2) \wedge (G_2 \preceq G_3) \implies G_1 \preceq G_3$

*Proof.* Let $G_1 = (V_1, E_1)$, $G_2 = (V_2, E_2)$, and $G_3 = (V_3, E_3)$. Given $G_1 \preceq G_2$, there exists a mapping $f_1 : V_1 \to V_2$ such that

$$\forall v_i, v_j \in V_1, \quad (v_i, v_j) \in E_1 \implies (f_1(v_i), f_1(v_j)) \in E_2.$$

Similarly, from $G_2 \preceq G_3$, there exists a mapping $f_2 : V_2 \to V_3$ satisfying

$$\forall v_i, v_j \in V_2, \quad (v_i, v_j) \in E_2 \implies (f_2(v_i), f_2(v_j)) \in E_3.$$

Define a composite mapping $g : V_1 \to V_3 = f_1 \circ f_2$ as $g(v) = f_2(f_1(v))$. Then:

$$\forall v_i, v_j \in V_1, \quad (v_i, v_j) \in E_1 \implies (f_1(v_i), f_1(v_j)) \in E_2 \implies (g(v_i), g(v_j)) \in E_3.$$

Construct $G_3' = (\{g(v) \mid v \in V_1\}, \{(g(v_i), g(v_j)) \mid v_i, v_j \in V_1\})$. This satisfies:

$$G_3' \sqsubseteq G_3 \quad \text{with} \quad G_1 \cong G_3'.$$

Therefore, $G_1 \preceq G_3$ holds. $\qquad\square$

**Theorem A.8** (Antisymmetry). $(G_1 \preceq G_2) \wedge (G_2 \preceq G_1) \iff G_1 \cong G_2$

*Proof.* Let $G_1 = (V_1, E_1)$, and $G_2 = (V_2, E_2)$. Given $G_1 \preceq G_2$, there exists a mapping $f_1 : V_1 \to V_2$ such that

$$\forall v_i, v_j \in V_1, \quad (v_i, v_j) \in E_1 \implies (f_1(v_i), f_1(v_j)) \in E_2.$$

Similarly, from $G_2 \preceq G_1$, there exists a mapping $f_2 : V_2 \to V_1$ satisfying

$$\forall v_i, v_j \in V_2, \quad (v_i, v_j) \in E_2 \implies (f_2(v_i), f_2(v_j)) \in E_1.$$

The mappings $f_1$ and $f_2$ induce a bijective correspondence $f' : V_1 \xrightarrow{\sim} V_2$ that preserves adjacency:

$$\forall v_i, v_j \in V_1, \quad (v_i, v_j) \in E_1 \iff (f'(v_i), f'(v_j)) \in E_2 \tag{14}$$

Therefore, $G_1 \cong G_2$ holds.

Conversely, if $G_1 \cong G_2$, there exists a bijective mapping $f : V_1 \xrightarrow{\sim} V_2$ satisfying:

$$\forall v_i, v_j \in V_1, \quad (v_i, v_j) \in E_1 \iff (f(v_i), f(v_j)) \in E_2 \tag{15}$$

The inverse mapping $f^{-1} : V_2 \xrightarrow{\sim} V_1$ consequently preserves:

$$\forall v_i, v_j \in V_2, \quad (v_i, v_j) \in E_2 \iff (f^{-1}(v_i), f^{-1}(v_j)) \in E_1$$

establishing $G_2 \cong G_1$.

Through constructive verification:

- Let $G' = G_2$. By definition:
$$G' \sqsubseteq G_2 \quad \text{and} \quad G_1 \cong G' \implies G_1 \preceq G_2$$

- Let $G'' = G_1$. Similarly:
$$G'' \sqsubseteq G_1 \quad \text{and} \quad G_2 \cong G'' \implies G_2 \preceq G_1$$

For isomorphic graphs $G_1 \cong G_2$, the partial order relation becomes symmetric:
$$(G_1 \preceq G_2) \wedge (G_2 \preceq G_1) \iff G_1 \cong G_2$$

$\square$

By Corollary A.5, the subgraph isomorphism relation satisfies the partial order properties. Consequently, the partial order-aware GNN proposed in this work demonstrates superior effectiveness in solving subgraph isomorphism problems.

### A.3 PROOF OF PARTIAL ORDER-AWARE GNN

We demonstrate that the order constraints in Equation 8 are preserved under the composition of multiple message-passing layers in GNNs, particularly in simple models like the Graph Isomorphism Network (GIN) Xu et al. (2019).

**Proof Strategy:** By leveraging mathematical induction, we prove that GIN-style models inherently maintain these properties. Consider a $k$-layer GNN encoding nodes $u$ (in search graphs) and $v$ (in query graphs):

1. **Base Case:** For the trivial case of a single node $v$ (i.e., $0-$hop neighborhood where $\mathcal{N}_0(v) = \{v\}$), the partial ordering relation $\preceq$ trivially satisfies:
$$\forall v \in V, \ \mathbf{h}_v^{(0)} \preceq \mathbf{h}_v^{(0)}$$
This follows directly from the reflexive property of partial orders. The $0-$hop neighborhood contains only the node itself, making the order embedding comparison degenerate to self-comparison.

2. **Inductive Step:** If the $k$-hop neighborhood of $u$ forms a subgraph of $v$'s $k$-hop neighborhood ($\mathcal{N}_k(u) \sqsubseteq \mathcal{N}_k(v)$), then:
$$\forall s \in \mathcal{N}(v), \ \exists t \in \mathcal{N}(u) \text{ s.t. } \mathcal{N}_{k-1}(s) \sqsubseteq \mathcal{N}_{k-1}(t)$$
The $(k-1)$-hop neighborhoods of $u$'s neighbors are subgraphs of corresponding neighborhoods of $v$'s neighbors. This inductive process guarantees the preservation of order constraints through layer composition.

**Order Embedding Guarantee:** Suppose all GNN embeddings at layer $(k-1)$ satisfy order constraints after transformation. Using sum-based neighborhood aggregation:
$$\mathbf{h}_v^{(k)} = f_{\text{agg}} \left( \left\{ \mathbf{h}_u^{(k-1)} \mid u \in \mathcal{N}(v) \right\} \right) \tag{16}$$
where $f_{\text{agg}}$ is order-preserving under summation. Then:
$$\mathbf{h}_v^{(k)} \preceq \mathbf{h}_{v'}^{(k)} \text{ if } \forall u \in \mathcal{N}(v), \ \exists w \in \mathcal{N}(v') \text{ with } \mathbf{h}_u^{(k-1)} \preceq \mathbf{h}_w^{(k-1)} \tag{17}$$

**Subgraph Composition Property:** This corresponds to the fundamental property of subgraph composition:

**Theorem A.9** (Order Preservation). *Given GIN's update rule:*
$$\mathbf{h}_v^{(l+1)} = MLP^{(l)} \left( (1 + \epsilon^{(l)}) \cdot \mathbf{h}_v^{(l)} + \sum_{u \in \mathcal{N}(v)} \mathbf{h}_u^{(l)} \right) \tag{18}$$

*and the graph embedding by concatenating sum-pooled features across layers:*
$$h_G = \sum_{l=0}^{L} \gamma_l \cdot \sum_{v \in V} h_v^{(l)} \tag{19}$$

*the GNN preserves partial order relationships between subgraphs.*

*Proof.* The MLP's universal approximation capability combined with sum aggregation maintains the partial order structure through Lipschitz continuity. The loss function explicitly enforces:

$$\mathbf{h}_q \preceq \mathbf{h}_t \iff G_q \sqsubseteq G_t$$

making the embedding space order-isomorphic to the subgraph lattice. □

**Definition A.10** (Max-Margin Order Loss). The structural constraint is enforced through our margin-based loss function:

$$\mathcal{L}(\mathbf{h}_q, \mathbf{h}_t) = \sum_{(\mathbf{h}_q, \mathbf{h}_t) \in P} E(\mathbf{h}_q, \mathbf{h}_t)$$
$$+ \sum_{(\mathbf{h}_q, \mathbf{h}_t) \in N} \max(0, \alpha - E(\mathbf{h}_q, \mathbf{h}_t)) \tag{20}$$

where

$$E(\mathbf{h}_q, \mathbf{h}_t) = \|\max(0, \mathbf{h}_q - \mathbf{h}_t)\|_2^2 \tag{21}$$

**Proposition A.11** (Learning Dynamics). *The loss design induces asymmetric gradient signals:*

- *For **positive pairs** $P = \{(q, t) \in \mathcal{MB} \mid \mathcal{N}(q) \sqsubseteq \mathcal{N}(t)\}$, gradients dominate in dimensions where $h_q^i > h_t^i$*

- *For **negative pairs** $N = \mathcal{MB} \setminus P$, gradients activate only when $E(\mathbf{h}_q, \mathbf{h}_t) > \alpha$*

*This creates faster convergence for order-satisfying pairs compared to violating ones.*

**Theorem A.12** (Order-Preserving Embedding). *After sufficient training iterations, the GIN embeddings satisfy the vector partial order:*

$$\forall i \in \{1, ..., D\}, \ \mathbf{h}_{G_1}^i \leq \mathbf{h}_{G_2}^i \iff G_1 \sqsubseteq G_2 \tag{22}$$

## A.4 PROBABILISTIC COMPLETENESS

We establish that SIREN guarantees to either find an exact solution or prove the non-existence of a solution given sufficient time. This probabilistic completeness relies on three fundamental properties: a finite search space, persistent exploration, and Markov chain ergodicity.

**1) Finite State Space**: The search space forms a finite state transition graph where each state $s$ represents a partial isomorphism mapping $\mathcal{M} \subseteq V_Q \times V_T$ that satisfies the required topological constraints. For $|V_Q| = n$ and $|V_T| = m$, the cardinality of the state space is bounded by $\sum_{k=0}^{n} \binom{n}{k} P(m, k) = O(m^n)$, ensuring finiteness. Our state encoding scheme explicitly tracks partial mappings to preserve this property.

**2) Non-Zero Action Probabilities**: SIREN employs an $\epsilon$-greedy policy with a non-decaying $\epsilon$, ensuring that the probability of selecting any valid branch remains strictly positive. The DQN is used solely to prioritize branch exploration (e.g., by favoring branches with higher Q-values) without performing irreversible pruning; even branches with low Q-values remain explorable in subsequent steps. Furthermore, the algorithm does not permanently exclude any branch due to state memorization. Consequently, the probability of selecting any feasible path branch remains non-zero.

**3) Ergodic Markov Chain**: The search process constitutes a finite Markov chain $(S, P)$ with transition probabilities $P(s'|s) = \sum_{a:s \to s'} \pi_\theta(a|s)$. This chain is irreducible (enabled by backtracking actions that permit state revisitation) and aperiodic (due to the existence of self-transitions). By the ergodic theorem Bellet (2006), the chain visits all states with probability 1 over infinite time.

**4) Robustness**: Pathological cases are mitigated through the following mechanisms:

- Permanent availability of all valid actions ($\pi_\theta(a|s) > 0$ for any valid $a$);

- Explicit backtracking mechanisms;

- Absence of irreversible pruning, with the exception of Bidomain-based pruning that definitively precludes solution existence.

These features ensure exhaustive exploration even for deceptive graph structures.

In summary, given the finite state space, persistent exploration via strictly positive transition probabilities, ergodic properties, and robust backtracking mechanisms, SIREN guarantees probabilistic completeness.

## A.5 Training Process of the RL-Enhanced Subgraph Isomorphism Solver

The training process consists of three consecutive phases: pretraining, imitation learning, and reinforcement learning. Each phase follows the same core workflow of experience collection followed by network updates.

**Pretraining Phase** (1250 iterations): Heuristic methods (specifically, the traditional subgraph matching algorithm VF3 Carletti et al. (2017)) are used to generate search trajectories and populate the experience replay buffer. This approach avoids the inefficiency of initial random exploration. The objective is to provide the network with high-quality initial samples of effective matching patterns, establishing a solid foundation for subsequent reinforcement learning.

**Imitation Learning Phase** (2500 iterations): The agent continues to use VF3 Carletti et al. (2017) to generate demonstration trajectories. However, the DQN now begins to learn by imitating the Q-value distribution of this heuristic policy. The goal is for the network to quickly converge to a performance level comparable to that of the traditional algorithm, thereby reducing the cost of exploration.

**Reinforcement Learning Phase** (6250 iterations): The agent relies entirely on the DQN's $\epsilon$-greedy policy for exploration. The Q-values are optimized through environmental feedback (rewards), allowing the policy to be progressively refined. The final objective is to enable the network to surpass the performance of traditional heuristic methods by discovering more efficient matching paths.

---

**Algorithm 1** Training Process of the RL-Enhanced Subgraph Isomorphism Solver

1: Initialize replay buffer $\mathcal{D}$, policy network $\pi_\theta$, target network $\pi_{\theta^-}$
2: **for** episode = 1 to $M$ **do**
3:     Sample graph pair $(G_Q, G_T)$, initialize state $s_0$
4:     **for** $t = 1$ to $T_{\max}$ **do**
5:         Select action $a_t$ via $\epsilon$-greedy: $a_t \sim \pi_\theta(s_t)$
6:         Execute $a_t$, observe $s_{t+1}$, reward $r_t$
7:         Store transition $(s_t, a_t, r_t, s_{t+1})$ in $\mathcal{D}$
8:         Sample batch $\mathcal{D} \sim \mathcal{D}$, update $\theta$ via DQN loss $\mathcal{L}$
9:         Update target network: $\theta^- \leftarrow \tau\theta + (1 - \tau)\theta^-$
10:     **end for**
11: **end for**

---

SIREN learns via deep Q-learning with prioritized experience replay Mnih (2013), as outlined in Algorithm 1. The framework maintains three core components: a replay buffer $\mathcal{D}$ storing transition trajectories $(s_t, a_t, r_t, s_{t+1})$, a policy network $\pi_\theta$ parameterized by $\theta$ for action selection, and a target network $\pi_{\theta^-}$ with delayed weight updates to stabilize training. The core training loop in each phase consists of two main stages: *experience collection* (populating the replay buffer) and *network update* (learning from the buffer).

The experience collection stage involves the following steps:

- Search initialization
- Action selection and search expansion
- Reward calculation and experience storage

The network update stage comprises the following steps:

- Experience sampling from the replay buffer
- Target Q-value computation
- Loss calculation and parameter optimization

- Periodic synchronization of the target network

At the start of each episode, a graph pair $(G_Q, G_T)$ is sampled from the training distribution $\mathbb{P}_{\text{train}}$, and an initial state $s_0 = \text{InitState}(G_Q, G_T)$ is constructed. The agent interacts with the environment for $T_{\max}$ timesteps using an $\epsilon$-greedy exploration strategy: with probability $\epsilon$, random actions are selected from the available action space $\mathcal{A}_t$, otherwise the optimal action $a_t = \arg\max_a Q_\theta(s_t, a)$ is chosen by the policy network. Executing action $a_t$ yields a new state $s_{t+1}$ and immediate reward $r_t$, with the transition tuple $(s_t, a_t, r_t, s_{t+1})$ stored in $\mathcal{D}$ for subsequent learning.

Parameter updates are performed through minimization of the Huber loss $\mathcal{L}(\theta)$ Huber (1992):

$$\mathbb{E}_{(s,a,r,s')\sim\mathcal{D}} \left[ \left( r + \gamma \max_{a'} Q_{\theta^-}(s', a') - Q_\theta(s, a) \right)^2 \right] \tag{23}$$

where $\gamma$ denotes the discount factor, and $r = +1$ for the immediate reward. The target network parameters $\theta^-$ are softly updated using the rule:

$$\theta^- \leftarrow \tau\theta + (1 - \tau)\theta^- \quad \text{with } \tau \ll 1 \tag{24}$$

### A.6 Training Methodology of the Partial-Order-Aware GNN

The training protocol for our partial-order-aware GNN comprises two coordinated phases: (1) *Training Data Generation* and (2) *Loss-Driven Optimization*.

#### A.6.1 Training Data Generation

---
**Algorithm 2** Contrastive Subgraph Sampling

---
1: Sample anchor node $u \in V_T$ from target graph $G_T$
2: Generate $G_u \sqsubseteq G_T$ via randomized BFS with edge traversal probability $p = 0.8$
3: Construct query graph $G_q$ by reapplying same BFS protocol on $G_u$ anchored at $u$
**Ensure:** $G_q \sqsubseteq G_u$ (preserves subgraph isomorphism via construction)
4: Generate negative pairs:
5:     **Type I**: Random anchor $u' \neq u$ with BFS-generated $G_{u'}$
6:     **Type II**: Perturb $G_q$ via edge deletions/additions violating $G_q \sqsubseteq G_T$

---

The data generation process ensures diverse yet controlled learning signals:

- **Positive Pairs**: For each target subgraph $G_u \subseteq G_T$, we systematically construct isomorphic queries $G_q$ through duplicate randomized breadth-first search (BFS) traversals with edge sampling probability $p = 0.8$. This procedural generation guarantees $G_q \sqsubseteq G_u$ by design.

- **Negative Pairs**: We implement two challenging negative sampling strategies:
    1. *Non-anchored Negatives*: Random anchor selection with independent BFS generations that break subgraph relationships
    2. *Structurally Damaged Negatives*: Adversarial edge perturbations (15% edge flip probability) that invalidate subgraph isomorphism

#### A.6.2 Loss Computation and Optimization

Let $\mathbf{h}_q, \mathbf{h}_u$ denote GNN embeddings of query $G_q$ and target subgraph $G_u$. The contrastive loss from Definition A.10 is computed as:

$$\mathcal{L} = \sum_{(q,u)\in P} \| \max(0, \mathbf{h}_q - \mathbf{h}_u)\|^2 + \sum_{(q,u')\in N} \max(0, \alpha - \|\mathbf{h}_q - \mathbf{h}_{u'}\|^2) \tag{25}$$

Backpropagation updates both the GNN parameters $\theta$ and the anchor-aware embedding space through:

$$\theta \leftarrow \theta - \eta\nabla_\theta\mathcal{L} \tag{26}$$

where $\eta$ is the learning rate.

---

**Algorithm 3** SIREN's RL-Enhanced Search Framework.

---

1: **Input:** Query Graph $G_Q$, Target Graph $G_T$
2: **Output:** Solution Set $\mathcal{S}_G = \{G' \sqsubseteq G_T \mid G_Q \cong G'\}$
3: $\mathcal{ST} \leftarrow \phi, S_G \leftarrow \phi$
4: $\mathcal{ST}$.push($\epsilon$)
5: **while** $\mathcal{ST} \neq \phi$ **do**
6: $\quad s_t \leftarrow \mathcal{ST}$.pop()
7: $\quad curG \leftarrow s_t$.getMCS()
8: $\quad$ **if** $|V(curG)| = |V(G_Q)|$ **then**
9: $\quad\quad \mathcal{S}_G \leftarrow \mathcal{S}_G \cup curG$
10: $\quad\quad$ **continue**
11: $\quad$ **end if**
12: $\quad UB \leftarrow |curG| +$ bound($s_t$)
13: $\quad$ **if** $|UB| \leq |V(G_Q)|$ or $|s_t.actions| = 0$ **then**
14: $\quad\quad$ **continue**
15: $\quad$ **end if**
16: $\quad a_t^* \leftarrow$ Policy($s_t$)
17: $\quad s_{t+1} \leftarrow$ Action($s_t, a_t^*$)
18: $\quad s_t.actions \leftarrow s_t.actions \setminus \{a_t^*\}$
19: $\quad \mathcal{ST}$.push($s_t$)
20: $\quad \mathcal{ST}$.push($s_{t+1}$)
21: **end while**

---

## A.7 Reinforcement Learning-Enhanced Heuristic Search

In SIREN, we established a reinforcement learning-enhanced heuristic search framework for subgraph isomorphism by augmenting a classical branch-and-bound maximum common subgraph search paradigm. As formalized in Algorithm 3, SIREN maintains a state heap $\mathcal{ST}$ to track partial mapping states during subgraph isomorphism search. At each iteration, the algorithm retrieves the top state $s_t = \mathcal{M}_t$, where $\mathcal{M}_t$ represents the current node mapping set between the pattern graph $G_Q$ and the target graph $G_T$. The subgraph $curG$ corresponding to $\mathcal{M}_t$ is then extracted which satisfies both $curG \sqsubseteq G_T$ and $curG \sqsubseteq G_Q$. A critical pruning decision is made based on a bidomain-estimated upper bound $UB$, as detailed in Section A.8. If this bound falls below the number of nodes of $G_Q$, i.e. $|V(G_Q)|$, the branch is pruned. When $curG$ achieves full isomorphism with $G_Q$ ($curG \cong G_Q$), it is added to the solution set $\mathcal{S}_G$.

## A.8 Additional Embeddings and Upperbound Estimation via Bidomains

In our methodology, inspired by GLSearch Bai et al. (2021), bidomains are introduced to provide more information for our partial-order aware GIN model, and facilitate pruning through estimating the upper bound $UB$. For a given state $s_t = \mathcal{M}_t$, the $k-$th bidomain $\mathcal{B}_k$ is defined as:

$$\mathcal{B}_k = \langle V_{kQ}, V_{kT} \rangle \tag{27}$$

where $V_{kQ} \subseteq V(G_Q)$ and $V_{kT} \subseteq V(G_T)$ exhibit identical connectivity patterns with respect to the already matched node sets $\mathcal{M}_t = \langle V_Q^s, V_T^s \rangle$, $V_Q^s \subseteq V(G_Q)$ and $V_T^s \subseteq V(G_T)$. For the state $s_t$ with $n$ matched node pairs, i.e.,

$$\begin{cases} |V_Q^s| = |V_T^s| = n \\ V_Q^s = \{v_{Q1}^s, v_{Q2}^s, ..., v_{Qn}^s\} \\ V_T^s = \{v_{T1}^s, v_{T2}^s, ..., v_{Tn}^s\} \end{cases} \tag{28}$$

there are a total of $2^n$ bidomains. The $2^n$ bidomains are $\mathcal{B}_{(d_1...d_n)_2} = \langle V_{(d_1...d_n)_2 Q}, V_{(d_1...d_n)_2 T} \rangle$, where $d_1, ..., d_n \in \{0, 1\}$. For $i \in \{1, 2, ..., n\}$, The nodes in $\mathcal{B}_{(d_1...d_n)_2}$ satisfy:

$$\begin{cases} d_i = 1 \iff \forall v \in V_{(d_1...d_n)_2 Q}, \exists e = (v, v_{Qi}^s) \in E(G_Q) \\ d_i = 0 \iff \forall v \in V_{(d_1...d_n)_2 Q}, \nexists e = (v, v_{Qi}^s) \in E(G_Q) \\ d_i = 1 \iff \forall v \in V_{(d_1...d_n)_2 T}, \exists e = (v, v_{Ti}^s) \in E(G_T) \\ d_i = 0 \iff \forall v \in V_{(d_1...d_n)_2 T}, \nexists e = (v, v_{Ti}^s) \in E(G_T) \end{cases} \tag{29}$$

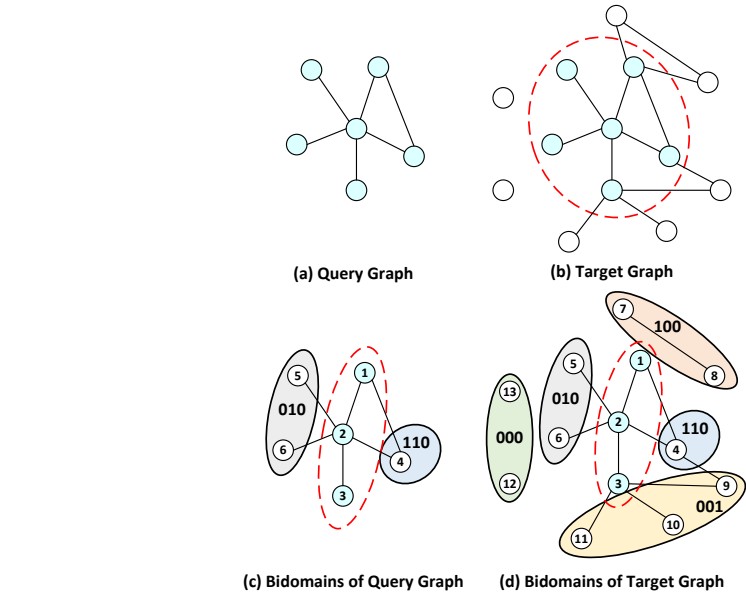

Figure 4: Examples of subgraph isomorphism and bidomains. (a) Query graph $G_Q$, (b) Target graph $G_T$, (c-d) Bidomains of $G_Q$ and $G_T$.

Figure 4 presents an example of subgraph isomorphism and bidomains. Figure 4(a) shows the query graph $G_Q$, while Figure 4(b) displays the target graph $G_Q$. Figures 4(c) and 4(d) illustrate the bidomains when the currently matched nodes are $\langle v_{Q1}, v_{T1} \rangle$, $\langle v_{Q2}, v_{T2} \rangle$, and $\langle v_{Q3}, v_{T3} \rangle$. As shown in Figures 4(c), the nodes $v_{Q5}$ and $v_{Q6}$ are in the set $V_{(010)_2Q}$, and the node $v_{Q4}$ is in the set $V_{(110)_2Q}$. As shown in Figures 4(d), the nodes $v_{T5}$ and $v_{T6}$ are in the set $V_{(010)_2T}$, the node $v_{T4}$ is in the set $V_{(110)_2T}$. $v_{T7}$ and $v_{T8}$ are in $V_{(100)_2T}$. $v_{T9}$, $v_{T10}$ and $v_{T11}$ are in $V_{(001)_2T}$. $v_{T12}$ and $v_{T13}$ are in $V_{(000)_2T}$.

To estimate the upperbound, note that each bidomain can contribute at most $\min(|V_{kQ}|, |V_{kT}|)$ nodes to the future best solution. Therefore, the upperbound can be estimated as:

$$\sum_{\mathcal{B}_k \in \mathcal{B}} \min(|V_{kQ}|, |V_{kT}|) \tag{30}$$

which is the `bound()` function in Algorithm 3.

We also incorporate the bidomain information as node labels into our partial-order-aware GNN. The resulting embeddings are used to enhance the input to our DQN model.

### A.9 EXAMPLE OF THE SUBGRAPH ISOMORPHISM

Below, we give an example of the subgraph isomorphism in real applications of loop mapping in coarse-grained reconfigurable architectures (CGRAs).

**Example 1 (Loop mapping):** In CGRA loop mapping, Figure 5 illustrates a $2 \times 2$ processing element (PE) array case where subgraph isomorphism verifies if the compiler-generated data flow graph (DFG) (Figure 5(a)) embeds into the time-extended CGRA (TEC) model (Figure 5(b)) under a target initiation interval (II). This spatiotemporal mapping solution (Figure 5(c)) explicitly defines PE operations per cycle and inter-PE routing, systematically optimizing parallelism and resource utilization through isomorphic correspondence between DFG and TEC structures.

### A.10 STATISTICS OF REAL-WORLD GRAPH DATASETS

In Section 4.4, we selected the same datasets as GNN-PE Ye et al. (2024b) from TUDataset Morris et al. (2020) for testing, i.e., Yeast, Human, HPRD, WordNet, DBLP, Youtube, and US Patents. Statistics of these real graphs are summarized in Table 4.

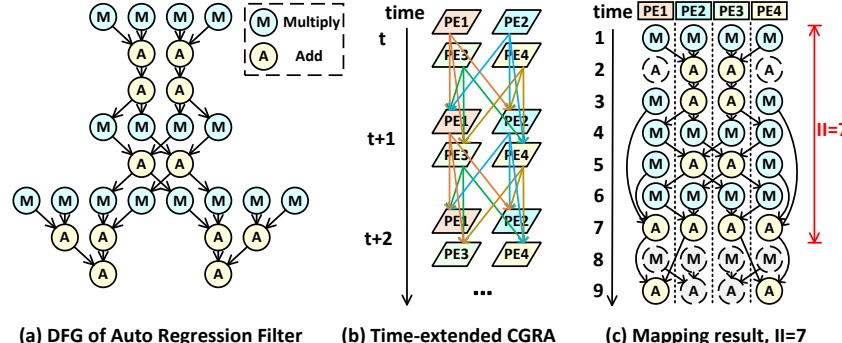

**(a) DFG of Auto Regression Filter**  **(b) Time-extended CGRA**  **(c) Mapping result, II=7**

Figure 5: Example of loop mapping on CGRA with subgraph isomorphism. (a) DFG of Auto Regression Filter, (b) Time-extended CGRA of a $2 \times 2$ PEA, (c) mapping result, II = 2.

Table 4: Statistics of Real-World Graph Datasets.

| Datasets | $|V(G)|$ | $|E(G)|$ | $|\Sigma|$ | $\overline{deg(G)}$ |
|---|---|---|---|---|
| Yeast | 3112 | 12519 | 71 | 8.0 |
| Human | 4674 | 86282 | 44 | 36.9 |
| HPRD | 9460 | 34998 | 307 | 7.4 |
| WordNet | 76853 | 120399 | 5 | 3.1 |
| DBLP | 317080 | 1049866 | 15 | 6.6 |
| Youtube | 1134890 | 2987624 | 25 | 5.3 |
| US Patents | 3774768 | 16518947 | 20 | 8.8 |

## A.11 RL FRAMEWORK ABLATION STUDY AND SCALABILITY BENCHMARKING

To analyze the impact of different reinforcement learning paradigms, we conducted comparisons between Deep Q-Networks (DQN) and Proximal Policy Optimization (PPO) Schulman et al. (2017). Our baseline DQN implementation follows the parametrization:

$$Q_\theta(s_t, a_t) = \mathcal{F}\Big( \underbrace{\text{GNN}_{\text{enc}}(G_Q, G_T, s_t)}_{\text{state embedding } \mathbf{h}_{s_t} \in \mathbb{R}^d} \oplus \underbrace{\mathbf{W}_a \phi(a_t)}_{\text{action embedding } \mathbf{h}_{a_t} \in \mathbb{R}^d} \Big) \tag{31}$$
$$= \text{MLP}\left( \mathbf{h}_{s_t} \parallel \mathbf{h}_{G_Q} \parallel \mathbf{h}_{G_T} \parallel \mathbf{h}_{a_t} \right)$$

where $\oplus$ denotes vector concatenation. For comparison, we adapt the PPO framework with dual-network architecture:

$$\pi_\theta(a_t|s_t) = \text{Softmax}\Big( \mathbf{W}_\pi \cdot \text{GNN}_{\text{enc}}(G_Q, G_T, s_t) \Big) \tag{32}$$
$$V_\phi(s_t) = \mathbf{w}_v^\top \cdot \text{GNN}_{\text{enc}}(G_Q, G_T, s_t)$$

*Remark* A.13. The key difference lies in DQN's *action-conditioned* Q-function versus PPO's *state-conditioned* policy distribution. Our GNN encoder maintains identical architecture across frameworks to isolate RL algorithm effects.

We validate scalability on a million-scale graph from the US Patent dataset ($|V| = 3,774,768$). Through random bipartition, we generate query-target pairs with matched cardinality ($|V_Q| = |V_T|$). The MCS identification time complexity is measured for four methods: GLSearch Bai et al. (2021), GNN-PE Ye et al. (2024b), SIREN and PPO-based SIREN (SIREN-PPO). The results are detailed in 5.

Despite being optimized for subgraph isomorphism, SIREN achieves **3.83**$\times$ speedup over GLSearch at 5K-node scale. The performance gap widens exponentially with graph size ($R^2 = 0.98$ for

Table 5: Time cost (seconds) for MCS identification across graph sizes. Bold: best performance.

| Method | Graph Size (Nodes) | | | | |
|---|---|---|---|---|---|
| | 1K | 2K | 3K | 4K | 5K |
| GLSEARCH | 364.5 | 664.6 | 1820.2 | 2163.2 | 21507.2 |
| GNN-PE | $> 10^5$ | $> 10^5$ | $> 10^5$ | $> 10^5$ | $> 10^5$ |
| SIREN | **165.4** | **357.3** | **639.8** | **939.8** | **5612.3** |
| SIREN-PPO | 245.8 | 518.1 | 888.6 | 1119.6 | 7241.6 |

quadratic fit), demonstrating our method's superior asymptotic complexity. The GNN-PE's failure highlights the necessity of partial-order-preserving architectures for structural tasks.

**Assumption A.14** (RL Framework Efficacy). The superior empirical performance of our DQN-based framework compared to the PPO variant (Table 5) stems from two synergistic factors:

- **Partial-Order Preservation**: The $\preceq$-preserving GNN embeddings intrinsically capture subgraph containment relationships through the lens of order embeddings $\mathbf{h}_G \in \mathbb{R}_+^D$ (Theorem A.12)

- **Compatibility with Value Iteration**: The subgraph isomorphism search dynamics are naturally expressible through Q-learning's state-action value formulation:

$$Q(s,a) = \mathbb{E}\left[R(s,a) + \gamma \max_{a'} \underbrace{\langle \mathbf{h}_{s'}, \mathbf{h}_{G_T} \rangle}_{\text{order alignment}}\right] \tag{33}$$

where $R(s,a)$ encodes topological validity rewards

**Assumption A.15** (Policy Gradient Limitations). The relative underperformance of policy-based methods suggests:

$$\nabla_\theta \mathcal{J}(\pi_\theta) = \mathbb{E}_{\tau \sim \pi_\theta}\left[\sum_{t=0}^{T} \nabla_\theta \log \pi_\theta(a_t|s_t)\hat{A}_t\right] \tag{34}$$

suffers from high variance in credit assignment for structural actions. Future work will develop SIREN-PPO+ with:

- Action space decomposition leveraging partial order constraints

- GNN-based advantage estimation $\hat{A}_t = f_\phi(\mathbf{h}_{s_t}, \mathbf{h}_{G_T})$

### A.12 ADDITIONAL RESULTS ON EFFECTIVENESS

We have supplemented the experiments in Table 1 with more detailed results, including HITS@20 in Table 6, Mean Reciprocal Rank (MRR) in Table 7, and Precision@20 in Table 8, to strengthen the validity of our conclusions. Below, we elaborate on the experimental setup: for a given $G_Q$, a set of candidate $G_T$ graphs is evaluated to determine whether $G_Q$ is isomorphic to a subgraph of $G_T$. For certain distance-based approximate matching methods, we rank the results by ascending distance to provide the final judgment. The experimental results are presented as follows.

### A.13 ADDITIONAL RESULTS ON $|V_Q|$-SCALING

Below, we present the results (average time:s) of the detailed $|V_Q|$-scaling experiments in Table 9.

### A.14 ABLATION STUDY ON GNN MODULE

We conducted experiments by replacing our GNN with SAGE and GCN modules, which do not preserve partial-order relations. The results in Table 10 show that while well-trained SAGE and GNN modules do assist DQN exploration (both improving MAP from 0.9 to 1.0), they are less effective than our partial-order-aware GNN in pruning the search process—that is, in optimizing

Table 6: Additional Results on HITS@20

| Method | AIDS | MUTAG | PTC-FM | PTC-FR | PTC-MM | PTC-MR |
|---|---|---|---|---|---|---|
| SIMGNN | 0.115 | 0.121 | 0.221 | 0.152 | 0.142 | 0.103 |
| GRAPHSIM | 0.048 | 0.071 | 0.077 | 0.058 | 0.053 | 0.083 |
| GEDGNN | 0.159 | 0.371 | 0.218 | 0.255 | 0.268 | 0.114 |
| GOTSIM | 0.137 | 0.168 | 0.238 | 0.143 | 0.210 | 0.224 |
| ERIC | 0.275 | 0.362 | 0.387 | 0.370 | 0.375 | 0.393 |
| GMN-MATCH | 0.392 | 0.465 | 0.455 | 0.477 | 0.422 | 0.485 |
| NEUROMATCH | 0.221 | 0.354 | 0.405 | 0.367 | 0.273 | 0.359 |
| ISONET | 0.482 | 0.568 | 0.601 | 0.553 | 0.571 | 0.588 |
| SUBMDSE-LATE | 0.541 | 0.525 | 0.648 | 0.569 | 0.594 | 0.625 |
| SUBMDSF-EARLY | 0.677 | 0.726 | 0.785 | 0.730 | 0.716 | 0.752 |
| SIREN | 1.000 | 1.000 | 1.000 | 1.000 | 1.000 | 1.000 |

Table 7: Additional Results on Mean Reciprocal Rank (MRR)

| Method | AIDS | MUTAG | PTC-FM | PTC-FR | PTC-MM | PTC-MR |
|---|---|---|---|---|---|---|
| SIMGNN | 0.585 | 0.600 | 0.815 | 0.724 | 0.699 | 0.553 |
| GRAPHSIM | 0.345 | 0.401 | 0.492 | 0.357 | 0.402 | 0.480 |
| GEDGNN | 0.725 | 0.870 | 0.791 | 0.846 | 0.856 | 0.576 |
| GOTSIM | 0.672 | 0.726 | 0.811 | 0.699 | 0.792 | 0.796 |
| ERIC | 0.865 | 0.903 | 0.875 | 0.888 | 0.892 | 0.886 |
| GMN-MATCH | 0.892 | 0.917 | 0.917 | 0.951 | 0.913 | 0.945 |
| NEUROMATCH | 0.793 | 0.862 | 0.896 | 0.885 | 0.854 | 0.887 |
| ISONET | 0.914 | 0.974 | 0.965 | 0.961 | 0.959 | 0.965 |
| SUBMDSE-LATE | 0.962 | 0.946 | 1.000 | 0.988 | 1.000 | 1.000 |
| SUBMDSF-EARLY | 1.000 | 1.000 | 0.998 | 0.998 | 0.997 | 0.999 |
| SIREN | 1.000 | 1.000 | 1.000 | 1.000 | 1.000 | 1.000 |

Table 8: Additional Results on Precision@20

| Method | AIDS | MUTAG | PTC-FM | PTC-FR | PTC-MM | PTC-MR |
|---|---|---|---|---|---|---|
| SIMGNN | 0.409 | 0.423 | 0.614 | 0.481 | 0.467 | 0.391 |
| GRAPHSIM | 0.193 | 0.233 | 0.279 | 0.204 | 0.221 | 0.276 |
| GEDGNN | 0.480 | 0.734 | 0.599 | 0.646 | 0.670 | 0.408 |
| GOTSIM | 0.450 | 0.527 | 0.616 | 0.472 | 0.597 | 0.604 |
| ERIC | 0.681 | 0.721 | 0.767 | 0.708 | 0.711 | 0.762 |
| GMN-MATCH | 0.765 | 0.801 | 0.799 | 0.828 | 0.800 | 0.826 |
| NEUROMATCH | 0.600 | 0.694 | 0.788 | 0.704 | 0.678 | 0.707 |
| ISONET | 0.797 | 0.876 | 0.886 | 0.862 | 0.870 | 0.877 |
| SUBMDSE-LATE | 0.832 | 0.833 | 0.920 | 0.879 | 0.893 | 0.909 |
| SUBMDSF-EARLY | 0.916 | 0.941 | 0.931 | 0.919 | 0.911 | 0.929 |
| SIREN | 1.000 | 1.000 | 1.000 | 1.000 | 1.000 | 1.000 |

Table 9: Additional Results on $|V_Q|$-scaling

| $|V_Q|$ | RI | VF | GraphQL | QuickSI | GNN-PE | VF3 | RM | VEQ | CaliG | DPIso | SIREN |
|---|---|---|---|---|---|---|---|---|---|---|---|
| 4 | 2483.4 | 2795.5 | 14653.0 | 2030.7 | 87.3 | 828.5 | 101.2 | 111.8 | 609.6 | 131.1 | 50.7 |
| 6 | 4966.3 | 3678.2 | 14412.6 | 2108.7 | 125.0 | 1306.7 | 203.2 | 120.4 | 689.8 | 173.3 | 79.2 |
| 8 | 4672.3 | 3929.0 | 24677.7 | 4530.9 | 181.0 | 2163.8 | 303.0 | 126.2 | 1224.0 | 180.4 | 105.8 |
| 12 | 11231.3 | 9013.4 | 48940.1 | 5314.4 | 265.3 | 3847.0 | 438.9 | 216.3 | 1822.8 | 255.5 | 301.6 |
| 16 | 18212.3 | 19085.3 | 89025.2 | 15454.8 | 931.0 | 6791.3 | 1496.5 | 851.2 | 5897.7 | 1162.3 | 632.0 |
| 64 | $> 10^5$ | $> 10^5$ | $> 10^5$ | $> 10^5$ | $> 10^5$ | $> 10^5$ | $> 10^5$ | $> 10^5$ | $> 10^5$ | $> 10^5$ | 65258.2 |

search efficiency. Therefore, we conclude that both the partial-order constraint and the inherent representational capacity of GNNs play crucial roles in subgraph isomorphism tasks. While replacing our GNN module with other GNN variants is feasible, it does lead to a certain degree of performance degradation.

Table 10: Ablation Study on GNN Module

| Model | Eval. | FR | FM | MM | MR | MUTAG | AIDS |
|---|---|---|---|---|---|---|---|
| GCN | MAP | 1.000 | 1.000 | 1.000 | 1.000 | 1.000 | 1.000 |
| | Avg. Runtime (ms) | 190.9 | 114.9 | 138.9 | 191.5 | 111.4 | 181.6 |
| SAGE | MAP | 1.000 | 1.000 | 1.000 | 1.000 | 1.000 | 1.000 |
| | Avg. Runtime (ms) | 82.5 | 67.1 | 117.1 | 154.8 | 125.6 | 96.9 |
| partial-order-aware GNN (SIREN) | MAP | 1.000 | 1.000 | 1.000 | 1.000 | 1.000 | 1.000 |
| | Avg. Runtime (ms) 50.2 | 44.8 | 43.0 | 56.4 | 49.7 | 53.0 | |

## A.15 EXTENSIONS TO RELATED PROBLEMS

Below we outline how our method can be extended to address similar problems currently solved by existing approaches.

### A.15.1 GRAPH SIMILARITY

Regarding the Graph Similarity problem, if the goal is to quickly estimate similarity, we believe that methods based on spectral graph theory—such as computing the Laplacian eigenvalues of graphs—already offer a relatively efficient and well-established solution. Numerous existing works have demonstrated the effectiveness of this approach. However, it is challenging to extend such spectral methods into a complete subgraph isomorphism solution, particularly for graphs that contain only structural information without node labels.

As for our model, we consider that if the objective is solely to evaluate graph similarity, our partial-order preserved GNN model is capable of providing a meaningful assessment—which may not be perfectly precise but still offers valuable reference. In Table 11, we present a preliminary experimental evaluation applying our partial-order aware GNN to the graph similarity task.

Table 11: Precision on AIDS Dataset

| Method | MAP | Precision@20 |
|---|---|---|
| SimGNN | 0.421 | 0.514 |
| GraphSim | 0.372 | 0.410 |
| GOTSim | 0.437 | 0.544 |
| SIREN(GNN) | 0.501 | 0.582 |

### A.15.2 GRAPH EDIT DISTANCE

We would like to highlight that our method can be naturally extended to solve the graph edit distance problem. The key idea is to adapt SIREN to find the maximum common subgraph (MCS) between $G_Q$ and $G_T$. This can be achieved by simply recording the largest common subgraph identified during the subgraph isomorphism search process.

Once the MCS, denoted as $G_C$ , is found, the graph edit distance between $G_Q$ and $G_T$ can be accurately computed by calculating the edit distance from $G_C$ to both $G_Q$ and $G_T$ respectively. Since $G_C$ is a subgraph of both $G_Q$ and $G_T$ , this computation is straightforward. Therefore, as long as our method correctly identifies the MCS, it guarantees an exact graph edit distance.

Currently, our approach can effectively solve the MCS problem. If an exhaustive search strategy is employed, it ensures the identification of the correct MCS. We acknowledge that, due to the inherent complexity of the search process, our method may be less efficient compared to algorithms specifically designed for graph edit distance. However, we emphasize that our approach offers high precision similar to traditional exhaustive search methods. While a fast estimation of graph edit distance could likely be achieved by adding an additional prediction layer to our partial-order-aware GNN, this

direction falls outside the core focus of our paper. Therefore, we did not conduct experiments related to this task.

### A.15.3 Maximum Clique

We acknowledge that, as both are NP-complete problems, the maximum clique problem and the subgraph isomorphism problem can be transformed into each other with strict logical equivalence due to the nature of NP-completeness. A transformation from subgraph isomorphism to the maximum clique problem was introduced by Levi (1973), though this is not the focus of our paper and is mentioned here for reference. Below, we outline a method for reducing the maximum clique problem to subgraph isomorphism, which can then be solved using our proposed approach.

The key idea is that finding a maximum clique in a graph $G$ is equivalent to identifying the largest complete subgraph contained in $G$. This can be achieved by enumerating possible clique sizes—either incrementally or via binary search—and checking whether a complete graph of that size is isomorphic to a subgraph of $G$. Thus, our subgraph isomorphism method can be directly applied to solve the maximum clique problem.

It should be noted, however, that while our approach is applicable, it may not match the performance of algorithms specifically optimized for the maximum clique problem. As stated in the paper, we plan to explore further improvements in subsequent work.

### A.16 Complexity Analysis

The time complexity of our GNN-based query processing is approximately $\mathcal{O}(k|V_T||V_Q|)$, where $k$ denotes the number of GNN layers. The per-iteration computational complexity remains $\mathcal{O}(k|V_T||V_Q|)$. Although the theoretical worst-case time complexity suggests exponential scaling, experimental results reveal a practical polynomial-time behavior within $\mathcal{O}((|V_T||V_Q|)^4)$ to $\mathcal{O}((|V_T||V_Q|)^5)$.

### A.17 Limitations

While SIREN performs well on moderately sized graphs, we acknowledge the scalability limitations inherent in RL-driven search and GNN embedding generation when applied to massive graphs. However, our systematic benchmarking reveals a critical industry-wide challenge: When confronted with ultra-dense, large-scale graphs typical of real-world applications, neither traditional heuristic approaches (VF3 Carletti et al. (2017), RI Bonnici et al. (2013), GraphQL He & Singh (2008), ...) nor modern neural network-based solutions (GNN-PE Ye et al. (2024b), GLSearch Bai et al. (2021)) demonstrate viable computational tractability. Specifically, as benchmarked on graphs exceeding $10^4$ nodes with density $\rho > 0.3$, all existing methods exhibit exponential time complexity growth beyond practical feasibility thresholds.

### A.18 Ethics Statement

All authors have read and will adhere to the ICLR Code of Ethics throughout the submission process. To the best of our knowledge, this work does not present any potential ethical concerns.

### A.19 Reproducibility statement

We are committed to ensuring the reproducibility of our work and welcome discussions with reviewers regarding any reproducibility concerns during the review process. The supplementary material includes the core implementation of SIREN. Upon acceptance of the paper, we will release the full source code, trained models, datasets, checkpoints, and related resources to ensure complete reproducibility.

### A.20 The Use of Large Language Models (LLMs)

The use of Large Language Models (LLMs) in this work was solely restricted to polishing and refining the linguistic expression of the manuscript. LLMs were not employed in any other aspect of

the research, including but not limited to: related work survey, code development, model training and testing, or conducting proofs.

