# OpenReview forum: "SIREN: Subgraph Isomorphism via Reinforcement Enhanced Graph Neural Networks"
_ICLR.cc/2026/Conference — Submitted to ICLR 2026_

### Official Review · Reviewer_8kgs · 2025-10-27

**Soundness:** 2
**Presentation:** 2
**Contribution:** 2
**Rating:** 4
**Confidence:** 3

**Summary:**

This paper presents SIREN, a novel framework that integrates a Deep Q-Network (DQN) with a partial order-aware Graph Neural Network (GNN) to solve the subgraph isomorphism problem. The key innovation lies in combining reinforcement learning to guide a tree-search process with GNN embeddings that preserve subgraph partial order relationships. The authors claim that SIREN achieves 100% precision (exact solutions) while being significantly faster than both state-of-the-art heuristic and learning-based methods.

**Strengths:**

1. The integration of DQN with a partial order-aware GNN for subgraph isomorphism is innovative. The idea of using RL to learn a node selection heuristic, replacing hand-crafted rules in traditional solvers, is a promising research direction.

2. The paper provides a theoretical discussion on probabilistic completeness, which is a crucial property for an exact solver and distinguishes it from many approximate learning-based methods.

**Weaknesses:**

1. The paper seems to omit key metrics. It reports only accuracy (not runtime) on trivial small graphs where SIREN may be slower than heuristic methods, and only runtime (not accuracy) on large graphs where its precision is unproven.

2. The method is designed and tested only on graphs with topological structure, ignoring the critical challenge of node and edge attributes. This severely limits its applicability to real-world problems where matching is defined by both structure and features (e.g., molecular graphs).

3. The bidomain-based pruning, while theoretically sound, incurs an exponential cost that could easily become a computational bottleneck on larger or sparser graphs, potentially negating its intended benefits.

**Questions:**

1. The training process involves three phases (pretraining, imitation, RL). How stable is this pipeline, and what is the relative contribution of each phase to the final performance? Is the RL phase essential, or does most of the performance come from imitating VF3?

2. The partial order-aware GNN is central to the node embeddings. How does its performance compare to a standard, off-the-shelf GNN model on this task? Is the partial order constraint the key factor, or is the gain primarily from the GNN's general representation power?

3. Given the complexity of the full SIREN pipeline, did you explore a simpler alternative? What is the evidence that the full RL-based search is necessary?

---

> ### Author Response · Authors · 2025-12-03
> **Response to Reviewer 8kgs---Part 1**
>
> We sincerely appreciate the reviewer's thoughtful analysis and constructive feedback on our work SIREN. We are particularly grateful for the recognition of our method's **innovative integration of DQN with a partial order-aware GNN** and **theoretical discussion on probabilistic completeness**. Below we address each concern systematically:
>
> > W1: The paper seems to omit key metrics. It reports only accuracy (not runtime) on trivial small graphs where SIREN may be slower than heuristic methods, and only runtime (not accuracy) on large graphs where its precision is unproven.
>
> We thank the reviewer for their comments regarding key evaluation metrics. In fact, our method already includes results on accuracy and runtime for trivial small graphs (see the last four rows of Table 1). As for the large graphs evaluated in Table 3, Figure 2, and Figure 3, all baselines compared in these experiments are exact solvers. Therefore, the accuracy of all methods is inherently 1.000, and this information is implicitly reflected in the experimental setup, which is why it was not explicitly reported. We have added corresponding clarifications in the revised version of the paper.
>
> ## Weakness 2
> > W2: The method is designed and tested only on graphs with topological structure, ignoring the critical challenge of node and edge attributes. This severely limits its applicability to real-world problems where matching is defined by both structure and features (e.g., molecular graphs).
>
> We thank the reviewer for raising the discussion on data with node and edge attributes. Our method is primarily designed for graphs without node labels, i.e., those containing only structural information, which represent a more challenging and information-sparse setting. Any additional label information inherently provides extra pruning cues that can benefit the search process. Indeed, for graphs that do include node and edge labels, both our method and conventional approaches such as VF3 and VEQ exhibit significantly reduced computation time, since label constraints help eliminate a large number of feasible solutions.Therefore, in simpler cases where node labels are available, even without additional training, the extra label information can assist our DQN in performing more effective pruning.
>
> Although retraining the GNN with labeled data could further enhance performance, our current model is already capable of handling such graphs without retraining. This is achieved by incorporating label-based constraints into the DQN exploration process. Since the GNN inherently integrates node label information, its structural pruning capability remains effective and satisfactory even without additional fine-tuning.
>
> **Results on Labelled graph**
> | Nodes of G_T | 1000 | 2000 | 3000 | 4000 | 10000 |
> | --- | --- | --- | --- | --- | --- |
> | Nodes of G_Q | 200 | 400 | 600 | 8000 | 2000 |
> | Edge Density | 0.4 | 0.3 | 0.2 | 0.2 | 0.2 |
> | VF3p | 272.4 | 624.0 | 183.4 | 6406.1 | 9859.3 |
> | VEQ | 109.2 | 147.0 | 133.0 | 3952.6 | 8687.2 |
> | SIREN | 87.8 | 247.1 | 118.7 | 2729.0 | 5485.7 |

---

> ### Author Response · Authors · 2025-12-03
> **Response to Reviewer 8kgs---Part 2**
>
> ## Weakness 3
> > W3: The bidomain-based pruning, while theoretically sound, incurs an exponential cost that could easily become a computational bottleneck on larger or sparser graphs, potentially negating its intended benefits.
>
> We thank the reviewer for raising the discussion on the complexity of the bidomain computation. Although the number of bidomains grows exponentially, the time complexity of computing them is actually constrained by the number of neighbors per node, making it sublinear in practice. In practice, the bidomain acts as a fast neighborhood filtering mechanism similar to a single-layer GNN, while being computationally lighter. For the specific implementation of bidomain-based pruning, please refer to the code provided in our supplementary materials.
>
> We provide simplified pseudocode to illustrate this implementation:
>
> ```
> Class Bidomain:
>     attrs: left, right, natts, bid
>     len: len(left) × len(right)
>
> Func get_natts_hash(node):
>     natts = [] if fuzzy_matching else config_node_feats
>     return tuple(node[n] for n in natts)
>
> Func unroll_bidomains(natts2bds):
>     return [bd for bds in natts2bds.values() for bd in bds]
>
> Func get_natts2g2abd_sg_nids(natts2g2nids, natts2bds, nn_map):
>     res = {}
>     sg1, sg2 = set(nn_map.keys()), set(nn_map.values())
>     for natts, g2nid in natts2g2nids:
>         left_cum, right_cum = set(), set()
>         if natts in natts2bds:
>             for bd in natts2bds[natts]:
>                 left_cum ∪= bd.left; right_cum ∪= bd.right
>         left_cum ∪= sg1 ∩ g2nid['g1']
>         right_cum ∪= sg2 ∩ g2nid['g2']
>         res[natts]['g1'], res[natts]['g2'] = left_cum, right_cum
>     return res
>
> Func assign_bids(natts2bds):
>     bid = 0
>     for natts in sorted(natts2bds.keys()):
>         for bd in natts2bds[natts]:
>             bd.bid = bid; bid += 1
> ```
>
> ## Question 1
> > Q1: The training process involves three phases (pretraining, imitation, RL). How stable is this pipeline, and what is the relative contribution of each phase to the final performance? Is the RL phase essential, or does most of the performance come from imitating VF3?
>
> We thank the reviewer for their comments regarding the stability of the reinforcement learning process and its functional role. We have conducted experiments using different random seeds for the RL training, and the results confirm that the MAP score remains consistently at 1.000, with only minor variations in runtime performance. This demonstrates the strong stability of our RL-based approach.
>
> | Seed | Eval. | FR | FM | MM | MR | MUTAG | AIDS |
> | --- | --- | --- | --- | --- | --- | --- | --- |
> | 0 | MAP | 1.000 | 1.000 | 1.000 | 1.000 | 1.000 | 1.000 |
> | | Avg. Runtime (ms) | 50.5 | 47.3 | 42.5 | 56.3 | 52.4 | 52.7 |
> | 100 | MAP | 1.000 | 1.000 | 1.000 | 1.000 | 1.000 | 1.000 |
> | | Avg. Runtime (ms) | 50.6 |  46.3 |  44.4 |  57.3 | 52.8 |  53.4 |
> | 200 | MAP | 1.000 | 1.000 | 1.000 | 1.000 | 1.000 | 1.000 |
> | | Avg. Runtime (ms) | 50.2 |  44.8 |  43.0 |  56.4 | 49.7 |  53.0 |
> | 300 | MAP | 1.000 | 1.000 | 1.000 | 1.000 | 1.000 | 1.000 |
> | | Avg. Runtime (ms) | 51.2 |  45.3 |  42.8 |  55.9 | 51.5 |  53.2 |
> | 400 | MAP | 1.000 | 1.000 | 1.000 | 1.000 | 1.000 | 1.000 |
> | | Avg. Runtime (ms) | 49.9 |  44.3 |  39.4 |  56.0 | 48.9 |  51.6 |
>
> Regarding the role of each component in the RL process: although using RL alone (with limited training episodes) or relying solely on imitation of VF3 can achieve a MAP of 1.000 on the small datasets in Table 1, neither approach alone achieves performance comparable to traditional methods like VF3 or VEQ—as we discovered during DQN training. Initially, we attempted to explore the solution space using only the combination of DQN and GNN, without any dependency on labels or search paths generated by solvers like VF3. While feasible, this strategy resulted in considerably longer solving times. On the other hand, imitation learning based solely on VF3 undermines the embedding generation capability of our GNN, which is exactly what we aimed to improve through subsequent DQN+GNN joint training.
>
> We have evaluated the solving speed of our original model (without imitation learning) on complex datasets, and found it to be slower than methods such as VF3. This observation motivated our subsequent refinement, leading to the current version of our work.
>
> **Ablation Study on imitation learning**
> | Nodes of G_T | 1000 | 2000 | 3000 | 4000 | 10000 |
> | --- | --- | --- | --- | --- | --- |
> | Nodes of G_Q | 200 | 400 | 600 | 8000 | 2000 |
> | Edge Density | 0.4 | 0.3 | 0.2 | 0.2 | 0.2 |
> | Without imitation | 7622.5 | 12320.1 | 7419.8 | $>10^5$ | $>10^5$ |
> | Only imitation | 17245.6 | 17969.5 | 10498.3 | $>10^5$ | $>10^5$ |
> | SIREN | 642.4 | 943.2 | 486.7 | 16842.4 | 34719.5|

---

> > ### Author Response · Authors · 2025-12-03
> > **Response to Reviewer 8kgs---Part 3**
> >
> > ## Question 2
> > > Q2: The partial order-aware GNN is central to the node embeddings. How does its performance compare to a standard, off-the-shelf GNN model on this task? Is the partial order constraint the key factor, or is the gain primarily from the GNN's general representation power?
> >
> > We thank the reviewer for their discussion on the role of the partial-order-aware GNN module. We conducted experiments by replacing our GNN with SAGE and GCN modules, which do not preserve partial-order relations. The results show that while well-trained SAGE and GNN modules do assist DQN exploration (both improving MAP from ~0.9 to 1.0), they are less effective than our partial-order-aware GNN in pruning the search process—that is, in optimizing search efficiency. Therefore, we conclude that both the partial-order constraint and the inherent representational capacity of GNNs play crucial roles in subgraph isomorphism tasks. While replacing our GNN module with other GNN variants is feasible, it does lead to a certain degree of performance degradation.
> >
> > | Model | Eval. | FR | FM | MM | MR | MUTAG | AIDS |
> > | --- | --- | --- | --- | --- | --- | --- | --- |
> > | GCN | MAP | 1.000 | 1.000 | 1.000 | 1.000 | 1.000 | 1.000 |
> > | | Avg. Runtime (ms) | 190.9 | 114.9 | 138.9 | 191.5 | 111.4 | 181.6 |
> > | SAGE | MAP | 1.000 | 1.000 | 1.000 | 1.000 | 1.000 | 1.000 |
> > | | Avg. Runtime (ms) | 82.5 | 67.1 | 117.1 | 154.8 | 125.6 | 96.9 |
> > | partial-order-aware GNN (SIREN) | MAP | 1.000 | 1.000 | 1.000 | 1.000 | 1.000 | 1.000 |
> > | | Avg. Runtime (ms) | 50.2 |  44.8 |  43.0 |  56.4 | 49.7 |  53.0 |
> > | 300 | MAP | 1.000 | 1.000 | 1.000 | 1.000 | 1.000 | 1.000 |
> >
> > ## Question 3
> > > Q3: Given the complexity of the full SIREN pipeline, did you explore a simpler alternative? What is the evidence that the full RL-based search is necessary?
> >
> > We thank the reviewer for the discussion regarding the role of the RL module. If the goal is to provide an approximate matching result, the RL module can indeed be omitted (as shown by the SIREN-GNN results in Table 2). Existing methods such as SubMDSE and IsoNet have also demonstrated strong performance in fuzzy matching. However, if an exact solution to subgraph isomorphism is required, an RL module—or at least a greedy brute-force search component—is necessary.
> >
> > We previously attempted to combine a simple greedy search module with our GNN, but struggled to effectively integrate the search process with the graph embeddings generated by the GNN. Our experiments showed that when using conventional search strategies, the pruning techniques in methods like VEQ and DPIso are already highly optimized. Introducing GNN-based guidance introduced additional computational overhead, and the greedy heuristic often led to inaccuracies, resulting in significantly increased search costs. Our test results indicate that the GNN+greedy search approach—possibly due to limitations in our current implementation—consumes significantly more time ($> 10 \times $) than methods like VF3, rendering such overhead unacceptable in practice. This made the combined approach substantially inferior to established methods like VEQ, which motivated our exploration of a DQN-based framework.
> >
> > In future work, we plan to explore ways to simplify the training process and reduce the complexity of the neural components where possible.
> >
> > If the reviewers have **any remaining concerns** that have not been addressed, or if there are **any other suggestions** to further improve the paper, **please feel free to inform us at any time**.
> >
> > Thank you again for your thoughtful feedback and the valuable discussions.
> >
> > Best regards,
> >
> > The authors of Paper 17918

---

### Official Review · Reviewer_TGyc · 2025-10-31

**Soundness:** 2
**Presentation:** 3
**Contribution:** 3
**Rating:** 4
**Confidence:** 4

**Summary:**

This paper proposes SIREN, combining Deep Q-Networks with partial order-aware GNNs to solve the subgraph isomorphism problem. The method uses DQN to learn node selection heuristics while maintaining provable completeness through bidomain-based pruning. The authors claim 100% precision on approximate matching tasks with modest computational time and speedups of ~52% over AI methods and ~21% over heuristics.

**Strengths:**

(1) The "partial order-aware" GNN pretraining is a clever and valuable contribution. Using a max-margin loss to enforce the geometric partial order is an elegant way to capture subgraph structure.
(2) AI + Search Framework: The overall approach of integrating a learned (DQN) policy with a formal, complete search algorithm (enhanced with bidomain pruning) is a strong and practical direction for tackling NP-complete combinatorial problems.
(3) The reported results are impressive. Achieving 100% MAP (Table 1) against approximate methods validates the "exact" nature of the solver. Furthermore, the claimed speedups against a wide range of strong baselines (Figure 2, Table 3), including GNN-PE, VEQ, and parallel methods like VF3p, suggest that the learned search policy is highly effective.

**Weaknesses:**

(1) Contribution 2 says explicitly that “a partial-order-aware GNN pretraining strategy” "eliminates dependency on solver-generated labels." While this is true for the GNN pretraining (Section 3.2), the DQN training (Appendix A.5) is heavily dependent on a traditional solver. The training process involves a 3-stage curriculum (Pretraining, Imitation Learning, RL) where the first two stages use VF3 to generate "high-quality initial samples" and "demonstration trajectories." This re-introduces the very solver bottleneck the paper claims to eliminate and makes the novelty of Contribution 2 feel less impactful on the entire system.
(2) The paper says “provable completeness” but what is actually proved is probabilistic completeness under a non-decaying ε-greedy exploration and with the assumption that the search can run long enough. That is not the same as the deterministically complete, bounded-depth, fully explored search in classic subgraph solvers. There is no rigorous proof that the ε-greedy DQN policy will visit all states in finite time.
(3) Code is promised “upon acceptance,” so current results cannot be verified.
(4) In Section 4.2, “In SIREN, we utilize 8 layers of Graph Isomorphism Networks (GIN) Xu et al. (2019) each with 64 dimensions for the embeddings.” However, from Table 1, it seems that all baselines use 3-5 layers. It is unclear how baselines perform with more parameters in this paper.

**Questions:**

(1) Given that SIREN also fails on 10K+ node dense graphs, how is this a fundamental improvement over baselines that fail on the same instances?
(2) Since GIN is WL-power-equivalent, how do you handle cases where WL cannot distinguish candidate nodes but SI still requires distinguishing them?

---

> ### Author Response · Authors · 2025-12-03
> **Response to Reviewer TGyc---Part 1**
>
> We sincerely appreciate the reviewer's thoughtful analysis and constructive feedback on our work SIREN. We are particularly grateful for the recognition of our method's **clever and valuable contribution of the elegant partial order-aware GNN**, **AI + Search Framework** and **impressive results**. Below we address each concern systematically:
>
> ## Weakness 1
> > W1: Contribution 2 says explicitly that “a partial-order-aware GNN pretraining strategy” "eliminates dependency on solver-generated labels." While this is true for the GNN pretraining (Section 3.2), the DQN training (Appendix A.5) is heavily dependent on a traditional solver. The training process involves a 3-stage curriculum (Pretraining, Imitation Learning, RL) where the first two stages use VF3 to generate "high-quality initial samples" and "demonstration trajectories." This re-introduces the very solver bottleneck the paper claims to eliminate and makes the novelty of Contribution 2 feel less impactful on the entire system.
>
> We thank the reviewer for their comments regarding Contribution 2. We acknowledge that during the training of our DQN, we incorporated imitation learning (based on VF3) to improve the subsequent search process—this was introduced as part of the DQN refinement. However, our original implementation does not inherently rely on paths or results from VF3, and thus still aligns with the contribution statement that a partial-order-aware GNN pretraining strategy helps avoid dependency on solver-generated labels.
>
> Even without any solver involvement, our framework remains valid and would not affect the MAP score of 1.000 reported in Table 1. That said, the DQN model trained without such guidance may exhibit slower convergence on certain adversarial or large-scale datasets compared to the current training strategy. To prevent any potential overclaim, we have decided to remove the phrase "eliminates dependency on solver-generated labels" from the contribution statement.
>
> ## Weakness 2
> > W2: The paper says “provable completeness” but what is actually proved is probabilistic completeness under a non-decaying ε-greedy exploration and with the assumption that the search can run long enough. That is not the same as the deterministically complete, bounded-depth, fully explored search in classic subgraph solvers. There is no rigorous proof that the ε-greedy DQN policy will visit all states in finite time.
>
> We thank the reviewer for raising the discussion on completeness. As clarified in Appendix A.4, our method is in fact probabilistically complete rather than deterministically complete. We have revised all references to "completeness" in the paper to **"probabilistic completeness"** to avoid overclaim.
>
> In our current implementation, we only incorporate masks of previously visited node pairs at each search state to prevent redundant additions. Under this design, the additional space complexity is O(|V_Q| + |V_T|), which does not guarantee deterministic completeness.
>
> Moreover, our approach can be easily modified to achieve deterministic completeness by maintaining, at each search node, a record of whether all feasible states have been fully explored. In that case, the space overhead would increase to O(|S| × (|V_Q| + |V_T|)), where |S| denotes the total number of visited states. This would lead to a linear growth in memory usage with the number of explored states. Considering scalability, we have chosen not to adopt this strategy. (In fact, our experiments indicate that such modification brings limited improvement in time efficiency.)
>
> A straightforward modification of this kind offers limited performance gains. Therefore, to achieve deterministic completeness, a more desirable approach would be to employ a hash-based method to store all fully explored states. However, this would also lead to a linear increase in memory usage as the search progresses. This represents a typical trade-off between space and time efficiency, which we intentionally avoid. To mitigate the issue of repeated exploration, we believe that employing a Q-value-based selection mechanism, such as Upper Confidence Bound (UCB), coupled with a specific incentive for encouraging exploration along less-visited paths, would be highly effective. This direction is one of our planned future improvements.

---

> ### Author Response · Authors · 2025-12-03
> **Response to Reviewer TGyc---Part 2**
>
> ## Weakness 3
> > W3: Code is promised “upon acceptance,” so current results cannot be verified.
>
> We thank the reviewer for their attention to reproducibility. We acknowledge that code and data availability are critical in this regard. As indicated in the checklist, upon acceptance we will **open-source all code, models, datasets, checkpoints, and resources​ necessary for full reproducibility**. At the time of submission, the full codebase and framework were not yet fully organized for public release; however, **the supplementary material already includes the core implementation of SIREN**. We believe the provided code is sufficient to reproduce our method, as our implementation and testing process does not involve any arbitrary or undocumented parameters (i.e., no "magic numbers").
>
> ## Weakness 4
> > W4: In Section 4.2, “In SIREN, we utilize 8 layers of Graph Isomorphism Networks (GIN) Xu et al. (2019) each with 64 dimensions for the embeddings.” However, from Table 1, it seems that all baselines use 3-5 layers. It is unclear how baselines perform with more parameters in this paper.
>
> We thank the reviewer for their attention to fair comparison. The discussion on the number of GNN layers is indeed one of the key points we emphasized in our experiments.
>
> As shown in Table 1 of the paper, we have already included experimental results using a GNN module containing only 3 layers with a hidden size of 10 (refer to the two rows of SIREN-Mini, which report both runtime and MAP results). The results demonstrate that SIREN-Mini, with only 3 GNN layers, is sufficient to guarantee perfect subgraph isomorphism decision accuracy (MAP = 1.000). However, the experiments also reveal that although the GNN inference in SIREN-Mini is significantly faster than that of the 8-layer GNN in SIREN, its overall efficiency in solving the complete subgraph isomorphism problem is comparable to SIREN. This is because the 3-layer GNN provides less accurate partial-order embeddings, resulting in less informative guidance for the subsequent search process. Furthermore, in large-scale graph evaluations, SIREN-Mini requires considerably more time than SIREN with its 8-layer GNN.
>
> ## Question 1
> > Q1: Given that SIREN also fails on 10K+ node dense graphs, how is this a fundamental improvement over baselines that fail on the same instances?
>
> We thank the reviewer for raising the important issue of scalability on very large and dense graphs. We would like to first restate the key contributions of our method:
>
> 1. We propose, for the first time, a hybrid DQN+GNN framework for subgraph isomorphism along with a partial-order-aware GNN module, supported by theoretical guarantees.
>
> 2. Compared to most neural approximate matching methods, our approach achieves 100% accuracy, while outperforming all existing exact solvers in terms of speed.
>
> When dealing with extremely large and dense graphs, it is true that all existing methods face challenges. However, our approach still possesses two distinct advantages in such settings:
>
> 1. Thanks to its GNN and DQN components, our method can be **naturally parallelized**. In contrast to many traditional solvers that lack native parallel support, this gives our approach significantly better scalability—especially when sufficient hardware resources are available, allowing computation power to be effectively translated into faster subgraph matching.
>
> 2. Our method is **highly memory-efficient**. Unlike traditional approaches such as GNN-PE and VEQ, it does not rely on precomputation of memory-intensive patterns. The additional memory overhead during exploration remains linear, greatly improving scalability on very large graphs.
>
> Therefore, we believe that for extremely large and dense graphs, our method shows stronger potential and offers a promising direction for the community.

---

> ### Author Response · Authors · 2025-12-03
> **Response to Reviewer TGyc---Part 3**
>
> ## Question 2
> > Q2: Since GIN is WL-power-equivalent, how do you handle cases where WL cannot distinguish candidate nodes but SI still requires distinguishing them?
>
> We thank the reviewer for raising the point regarding the Weisfeiler–Lehman (WL) test, which indeed serves as a key theoretical foundation of our approach. It is precisely because there exist cases where the WL test cannot distinguish certain candidate nodes (while subgraph isomorphism still requires discriminating between them), that we employ a DQN to exhaustively explore the entire solution space.
>
> In our framework, the GIN module is responsible only for generating graph embeddings that guide the DQN’s search process; it does not play a decisive role in the final solution. Although we have designed the GIN embeddings to preserve partial-order relationships as much as possible, the DQN retains the ability to explore alternative search paths even when the WL test fails to distinguish graphs.
>
> Intuitively, our approach relies on the DQN’s capacity to explore the solution space and its varying Q-value predictions to differentiate and examine candidate nodes, even when the WL test outputs are identical or the GIN produces similar graph embeddings.
>
> If the reviewers have **any remaining concerns** that have not been addressed, or if there are **any other suggestions** to further improve the paper, **please feel free to inform us at any time**.
>
> Thank you again for your thoughtful feedback and the valuable discussions.
>
> Best regards,
>
> The authors of Paper 17918

---

### Official Review · Reviewer_qaH3 · 2025-11-03

**Soundness:** 3
**Presentation:** 3
**Contribution:** 3
**Rating:** 4
**Confidence:** 5

**Summary:**

The paper proposes SIREN, a novel framework that integrates Graph Neural Networks (GNNs) with Deep Reinforcement Learning (DQN) to solve the subgraph isomorphism problem. The method addresses both existence determination and complete solution enumeration with provable completeness. SIREN uses a partial-order-aware GNN to encode graph embeddings and a DQN with bidomain-based pruning to guide the search process efficiently. Extensive experiments on real-world and synthetic datasets show that SIREN achieves 100% precision and outperforms both traditional heuristic methods and recent neural approaches in terms of speed and accuracy.

**Strengths:**

i)，Novel and Deep Methodological Integration: The paper's primary strength lies not in simply replacing parts of a traditional algorithm with AI, but in a deep architectural fusion of three distinct paradigms: 1，Reinforcement Learning as a Search Guide: A Deep Q-Network (DQN) is used to learn an optimal policy for selecting the next node pair during the tree search. This moves beyond hand-crafted heuristics to a data-driven, adaptive strategy. 2，Graph Neural Networks as State Representers: A custom GNN is employed to generate meaningful embeddings for the state (the current partial mapping and the graphs) and actions (candidate node pairs). This allows the RL agent to reason about graph structure in a continuous space.
ii), Theoretically Grounded and Self-Supervised Pretraining: The introduction of the partial-order-aware GNN is a significant contribution that addresses a major bottleneck in learning-based approaches. It explicitly models the subgraph isomorphism relationship as a partial order in the embedding space. The loss function is designed to ensure that if G1 is isomorphic to a subgraph of G2, then each dimension of G1's embedding is less than or equal to that of G2's.

**Weaknesses:**

i), Inherent Scalability Limitations of the Approach: While SIREN outperforms other methods, it does not overcome the fundamental NP-hard nature of the problem. The framework still relies on a tree search whose worst-case time complexity remains exponential. The experiments show that on "ultra-dense, large-scale graphs" (e.g., >10^4 nodes, density ρ>0.3), SIREN, like all other solvers, becomes impractically slow. The claim of "polynomial-time behavior" in practice is observed, but not proven, and is likely highly dependent on graph structure. The memory footprint could be a critical constraint. The use of GNNs for embedding generation and a DQN for action selection, especially with large action spaces that require partitioning, implies high GPU memory usage. This could limit its application on hardware with constrained resources.
ii), Complex and Computationally Expensive Training Pipeline: The three-phase training process (Pretraining, Imitation Learning, Reinforcement Learning) is a double-edged sword. It requires running a traditional solver (VF3) to generate initial demonstration data, which itself can be slow. The need to train multiple components (GNN and DQN) sequentially makes the overall training process complex, long, and computationally intensive. This high barrier to entry could hinder adoption and make hyperparameter tuning difficult.
iii), Narrow Focus and Unverified Generalizability: 1,The method is highly specialized for the exact subgraph isomorphism problem. Its performance on related but distinct tasks like graph similarity, graph edit distance, or inexact matching is not explored. 2, The promise to extend to other NP problems (like the maximum clique problem) remains future work. The architecture's dependency on the specific "bidomain" structure for subgraph isomorphism may not translate directly to other problem domains.

**Questions:**

1，How does SIREN handle graphs with noisy or missing labels? The current method assumes clean graph structures; robustness to noisy data is not evaluated.

2， Could the framework be adapted for inexact or fuzzy subgraph matching? The current focus is on exact isomorphism; extending to approximate matching would broaden applicability.

3， What is the performance of SIREN on graphs with heterogeneous node/edge types? The experiments primarily use homogeneous graphs; generalization to heterogeneous graphs is unclear.

4， The very recent and highly related work D2Match (Liu et al., ICML 2023) also addresses the subgraph matching problem by combining deep learning with theoretical degeneracy. Please provide a detailed comparison between SIREN and D2Match, highlighting your method's relative advantages and limitations in terms of methodological approach, theoretical guarantees, and computational complexity.

---

> ### Author Response · Authors · 2025-12-03
> **Response to Reviewer qaH3---Part 1**
>
> We sincerely appreciate the reviewer's thoughtful analysis and constructive feedback on our work SIREN. We are particularly grateful for the recognition of our method's **Novel and Deep Methodological Integration** and **Theoretically Grounded and Self-Supervised Pretraining**. Below we address each concern systematically:
>
> ## Weakness 1
> > W1: Inherent Scalability Limitations of the Approach: While SIREN outperforms other methods, it does not overcome the fundamental NP-hard nature of the problem. The framework still relies on a tree search whose worst-case time complexity remains exponential. The experiments show that on "ultra-dense, large-scale graphs" (e.g., >10^4 nodes, density ρ>0.3), SIREN, like all other solvers, becomes impractically slow. The claim of "polynomial-time behavior" in practice is observed, but not proven, and is likely highly dependent on graph structure. The memory footprint could be a critical constraint. The use of GNNs for embedding generation and a DQN for action selection, especially with large action spaces that require partitioning, implies high GPU memory usage. This could limit its application on hardware with constrained resources.
>
> **About NP-Hard Nature**
> We thank the reviewer for the discussion regarding the NP-hard nature of the problem. We acknowledge that our method does not overcome the fundamental NP-hard nature of the subgraph isomorphism problem. In fact, given that subgraph isomorphism is NP-complete, no existing method can fundamentally circumvent its computational complexity—they can only aim to improve practical efficiency, particularly on general or non-adversarial datasets. We believe that anyone who devises an efficient polynomial-time algorithm for this problem would effectively prove **P = NP**, thereby solving a famous Millennium Prize Problem and undoubtedly earning accolades such as the **Turing Award** and the **Fields Medal** :)
>
> Therefore, based on the current understanding of computational complexity, we must conclude that any method claiming to "solve" the NP-hard nature of subgraph isomorphism inherently involves an overclaim. In practice, such methods can only either provide approximate solutions through heuristic guesses within polynomial time—which naturally avoids long running times but cannot guarantee exact solutions (as seen in methods like D2Match and SubMDSE cited in our paper)—or offer exact solutions with a certain probability, still within polynomial time.
>
> **About Training Cost**
> We thank the reviewer for raising the discussion on training cost. We provide the total training duration of our final model in the table below, including both the pretrained GNN phase and DQN training period. For the DQN component, we trained for 10000 iterations per dataset group. Across the 18 dataset groups utilized in our experiments (FR, FM, MM, MR, MUTAG, AIDS, FM, NCI, MOLT, Yeast, Human, HPRD, WordNet, DBLP, Youtube, US Patents, adversarial dense dataset, and synthetic dataset), we provide the total training duration and average time per iteration below. As specified in Section 4.2, all training was conducted utilizing 4 $\times$ NVIDIA A800 GPUs.
>
> **Training Time**
> | Period | Epochs/Iterations | est. Time per epoch/iter | Total Time |
> | --- | --- | --- | --- |
> | GNN Pretraining | 50000 Epochs | 1.33s | \~66433s (\~18.4h) |
> | DQN Training | 180000 Iterations | 0.23s | \~41709s (\~11.6h) |
> | Total |  |  | \~108142s (\~30.0h) |
>
> As demonstrated in the preceding table, SIREN incurs manageable computational overhead during its full training cycle, requiring only approximately 30 GPU-hours. This resource utilization leverages commercially accessible hardware: 4 $\times$ NVIDIA A800 GPUs.
>
> In accordance with your request, we have also evaluated the inference cost of our method on large-scale graphs containing millions of nodes, along with the cost of our original training process. The accompanying figures present the GPU memory/time curves. A subset of the numerical results used to generate these curves is provided in the table below. The data clearly demonstrates that our method maintains low memory (Maximum cost ~ 10GB for ultra-large graph) and GPU overhead throughout, which is a notable advantage.
>
> **GPU Memory Usage (MB)**
> | Time | Test_Small(US Patents) | Test_Large(Synthetic) | Train |
> | --- | --- | --- | --- |
> | 0min | 573 | 763 | 437 |
> | 1min | 585 | 779 | 577 |
> | 2min | - | 803 | 601 |
> | 3min | - | 823 | 621 |
> | 4min | - | 1995 | 649 |
> | 5min | - | 1995 | 669 |
> | 6min | - | 1995 | 689 |
> | 7min | - | 2005 | 713 |
> | 8min | - | 2027 | 737 |
> | 9min | - | 9197 | 757 |
> | 10min | - | 9207 | 765 |
> | Maximum | 597 | 10244 | 2018 |
>
> We believe that, in contrast to existing approaches which often entail high migration costs (e.g., GNN-PE with its expensive indexing overhead) or excessive memory consumption (e.g., VEQ), our method offers a highly favorable balance of training cost and overall efficiency, coupled with strong transferability.

---

> ### Author Response · Authors · 2025-12-03
> **Response to Reviewer qaH3---Part 2**
>
> ## Weakness 2
> > W2: Complex and Computationally Expensive Training Pipeline: The three-phase training process (Pretraining, Imitation Learning, Reinforcement Learning) is a double-edged sword. It requires running a traditional solver (VF3) to generate initial demonstration data, which itself can be slow. The need to train multiple components (GNN and DQN) sequentially makes the overall training process complex, long, and computationally intensive. This high barrier to entry could hinder adoption and make hyperparameter tuning difficult.
>
> We thank the reviewer for their attention to the training cost, which is indeed an important aspect of our method's practical application. As mentioned in our previous response, the total training time is approximately 30 hours on a previous-generation Nvidia A800 GPU. We believe this cost is reasonable—as long as the model is not trained on extremely large graphs, which is exactly what we aim to avoid by using multiple smaller datasets.
>
> Regarding the cost of generating search paths using methods like VF3, we note that a straightforward data augmentation strategy for subgraph isomorphism is to merge two $G_T$ graphs. By applying this approach, even though traditional methods may be slow on very large graphs, we can still guide the model's solution exploration by combining the search results of multiple smaller graph segments into a larger graph. This strategy plays a key role in reducing our overall training cost.
>
> ## Weakness 3
> > W3: Narrow Focus and Unverified Generalizability: 1,The method is highly specialized for the exact subgraph isomorphism problem. Its performance on related but distinct tasks like graph similarity, graph edit distance, or inexact matching is not explored. 2, The promise to extend to other NP problems (like the maximum clique problem) remains future work. The architecture's dependency on the specific "bidomain" structure for subgraph isomorphism may not translate directly to other problem domains.
>
> We thank the reviewer for their valuable feedback regarding the scalability and generalization of our method. Given the time constraints of the rebuttal period, it is challenging to fully develop and validate a generalized framework. Therefore, we will limit our response to a theoretical discussion on how our approach could be extended to address other types of problems. We have added the discussion in the appendix on extending our method to related problems.
>
> **1. Graph Similarity**
>
> Regarding the Graph Similarity problem, if the goal is to quickly estimate similarity, we believe that methods based on spectral graph theory—such as computing the Laplacian eigenvalues of graphs—already offer a relatively efficient and well-established solution. Numerous existing works have demonstrated the effectiveness of this approach. However, it is challenging to extend such spectral methods into a complete subgraph isomorphism solution, particularly for graphs that contain only structural information without node labels.
>
> As for our model, we consider that if the objective is solely to evaluate graph similarity, our partial-order preserved GNN model is capable of providing a meaningful assessment—which may not be perfectly precise but still offers valuable reference. Below, we present a preliminary experimental evaluation applying our partial-order aware GNN to the graph similarity task.
>
> **Precision on AIDS**
>
> | Method | MAP | Precision@20 |
> | --- | --- | --- |
> | SimGNN | 0.421 | 0.514 |
> | GraphSim | 0.372 | 0.410 |
> | GOTSim | 0.437 | 0.544 |
> | SIREN(GNN) | 0.501 | 0.582 |

---

> ### Author Response · Authors · 2025-12-03
> **Response to Reviewer qaH3---Part 3**
>
> **2. Graph Edit Distance**
>
> We would like to highlight that our method can be naturally extended to solve the graph edit distance problem. The key idea is to adapt SIREN to find the maximum common subgraph (MCS) between $G_Q$ and $G_T$. This can be achieved by simply recording the largest common subgraph identified during the subgraph isomorphism search process.
>
> Once the MCS, denoted as $G_C$ , is found, the graph edit distance between $G_Q$ and $G_T$ can be accurately computed by calculating the edit distance from $G_C$ to both $G_Q$ and $G_T$ respectively. Since $G_C$ is a subgraph of both $G_Q$ and $G_T$ , this computation is straightforward. Therefore, as long as our method correctly identifies the MCS, it guarantees an exact graph edit distance.
>
> Currently, our approach can effectively solve the MCS problem. If an exhaustive search strategy is employed, it ensures the identification of the correct MCS. We acknowledge that, due to the inherent complexity of the search process, our method may be less efficient compared to algorithms specifically designed for graph edit distance. However, we emphasize that our approach offers high precision similar to traditional exhaustive search methods. While a fast estimation of graph edit distance could likely be achieved by adding an additional prediction layer to our partial-order-aware GNN, this direction falls outside the core focus of our paper. Therefore, we did not conduct experiments related to this task.
>
> **3. Maximum Clique**
>
> We acknowledge that, as both are NP-complete problems, the maximum clique problem and the subgraph isomorphism problem can be transformed into each other with strict logical equivalence due to the nature of NP-completeness. A transformation from subgraph isomorphism to the maximum clique problem was introduced by Levi et al. [1], though this is not the focus of our paper and is mentioned here for reference. Below, we outline a method for reducing the maximum clique problem to subgraph isomorphism, which can then be solved using our proposed approach.
>
> The key idea is that finding a maximum clique in a graph $G$ is equivalent to identifying the largest complete subgraph contained in $G$. This can be achieved by enumerating possible clique sizes—either incrementally or via binary search—and checking whether a complete graph of that size is isomorphic to a subgraph of $G$. Thus, our subgraph isomorphism method can be directly applied to solve the maximum clique problem.
>
> It should be noted, however, that while our approach is applicable, it may not match the performance of algorithms specifically optimized for the maximum clique problem. As stated in the paper, we plan to explore further improvements in subsequent work.
>
> [1] G. Levi. A note on the derivation of maximal common subgraphs of two directed or undirected graphs. Calcolo, 9(4):341–352, 1973.
>
> ## Question 1
> > Q1: How does SIREN handle graphs with noisy or missing labels? The current method assumes clean graph structures; robustness to noisy data is not evaluated.
>
> We thank the reviewer for raising the discussion on noisy data. Our method is primarily designed for graphs without node labels, i.e., those containing only structural information, which represent a more challenging and information-sparse setting. Therefore, in simpler cases where node labels are available, even without additional training, the extra label information can assist our DQN in performing more effective pruning. As a result, factors such as label vocabularies, noise levels, or missing attributes do not adversely affect our approach. In fact, the presence of labels provides additional pruning cues that are inherently beneficial compared to the purely structural setting.
>
> **Results on Labelled graph**
> | Nodes of G_T | 1000 | 2000 | 3000 | 4000 | 10000 |
> | --- | --- | --- | --- | --- | --- |
> | Nodes of G_Q | 200 | 400 | 600 | 8000 | 2000 |
> | Edge Density | 0.4 | 0.3 | 0.2 | 0.2 | 0.2 |
> | VF3p | 272.4 | 624.0 | 183.4 | 6406.1 | 9859.3 |
> | VEQ | 109.2 | 147.0 | 133.0 | 3952.6 | 8687.2 |
> | SIREN | 87.8 | 247.1 | 118.7 | 2729.0 | 5485.7 |

---

> ### Author Response · Authors · 2025-12-03
> **Response to Reviewer qaH3---Part 4**
>
> ## Question 2
> > Q2: Could the framework be adapted for inexact or fuzzy subgraph matching? The current focus is on exact isomorphism; extending to approximate matching would broaden applicability.
>
> We thank the reviewer for raising the discussion on inexact or fuzzy subgraph matching. Our current framework supports inexact (fuzzy) subgraph matching through two distinct approaches. The first method is similar to the graph similarity-based prediction described in our response to W3, where the partial-order-aware GNN provides an approximate matching score. The corresponding accuracy results can be found in the "SIREN-GNN" row of Table 2 in the paper, which reflects the model's ability to predict approximate subgraph matching.
>
> The second approach yields an explicit inexact matching solution by computing the maximum common subgraph (MCS) within our framework. The unmatched portions are then identified as the differences between the input graphs and the MCS. This strategy aligns with the graph edit distance calculation method outlined in our response to W3.
>
> ## Question 3
> > Q3: What is the performance of SIREN on graphs with heterogeneous node/edge types? The experiments primarily use homogeneous graphs; generalization to heterogeneous graphs is unclear.
>
> As noted in our response to your Q1, our method is primarily designed for graphs that contain only structural information without node or edge labels. Any additional label information inherently provides extra pruning cues that can benefit the search process. Indeed, for graphs that do include node and edge labels, both our method and conventional approaches such as VF3 and VEQ exhibit significantly reduced computation time, since label constraints help eliminate a large number of feasible solutions.
>
> Although retraining the GNN with labeled data could further enhance performance, our current model is already capable of handling such graphs without retraining. This is achieved by incorporating label-based constraints into the DQN exploration process. Since the GNN inherently integrates node label information, its structural pruning capability remains effective and satisfactory even without additional fine-tuning.
>
> ## Question 4
> > Q4: The very recent and highly related work D2Match (Liu et al., ICML 2023) also addresses the subgraph matching problem by combining deep learning with theoretical degeneracy. Please provide a detailed comparison between SIREN and D2Match, highlighting your method's relative advantages and limitations in terms of methodological approach, theoretical guarantees, and computational complexity.
>
> D2Match is a subgraph matching approach that integrates deep learning with the concept of degeneracy. It theoretically reduces the NP-complete subgraph matching problem to a subtree matching task, which is then transformed into a bipartite perfect matching problem. By leveraging the tree-structured aggregation mechanism of Graph Neural Networks (GNNs), it achieves linear time complexity. Additionally, it can incorporate cycle structures—abstracted as super-nodes—and node attributes to enhance performance.
>
> While the mathematical formulation of D2Match is elegant, it remains an approximate matching method—that is, it trades off some computational time to produce a probabilistic subgraph isomorphism decision. In contrast, our method employs Deep Q-Networks (DQN) to exhaustively explore the solution space of subgraph isomorphism, while retaining the capability for fuzzy matching via partial-order-aware GNNs.
>
> Theoretically, D2Match has a complexity of $\mathcal{O}(|V_T|^2)$, which is the same as our GNN module and is significantly lower than the theoretical complexity of our method (exponential) as well as its empirically estimated complexity, which ranges from $\mathcal{O}((|V_T||V_Q|)^4)$ to $\mathcal{O}((|V_T||V_Q|)^5)$.
>
> Compared to our approach, D2Match exhibits stronger performance in fuzzy matching and offers faster execution. However, our method guarantees exact solutions to the subgraph isomorphism problem—a capability that D2Match does not possess.
>
> If the reviewers have **any remaining concerns** that have not been addressed, or if there are **any other suggestions** to further improve the paper, **please feel free to inform us at any time**.
>
> Thank you again for your thoughtful feedback and the valuable discussions.
>
> Best regards,
>
> The authors of Paper 17918

---

### Official Review · Reviewer_8b3e · 2025-11-07

**Soundness:** 3
**Presentation:** 3
**Contribution:** 3
**Rating:** 6
**Confidence:** 4

**Summary:**

This paper introduces SIREN, a framework that leverages a poset-aware GNN to produce structurally consistent graph/state embeddings, combined with a DQN-based policy to learn node-pair matching order and backtracking strategies. Together with bidomain-based upper bounds and pruning, SIREN aims to accelerate subgraph isomorphism search. The authors report 100% MAP on approximate matching tasks and significant speedups over heuristic solvers and GNN-PE on exact matching/enumeration benchmarks. The training pipeline consists of a large-scale supervised pretraining stage (50k epochs), followed by reinforcement learning fine-tuning.

**Strengths:**

1. Well-structured and modular design. The decomposition into state embedding, action embedding, upper-bound–guided pruning, and action-block evaluation is clear and reproducible. The interaction between action embedding ( $W_a$ ) and bidomain constraints in the Q-function is explicitly defined.
2. Comprehensive empirical evaluation across accuracy, efficiency, and scalability. The experiments cover approximate prediction, exact solving, and large/dense synthetic graphs.
3. Discussion of (conditional) completeness. The appendix uses the notion of probabilistic completeness, linking it to sufficient time and persistent exploration, which avoids portraying RL search as a black-box heuristic.

**Weaknesses:**

1. Ambiguity in the “completeness” claim.

    The main text repeatedly uses terms such as “provably complete” and “guarantees completeness,” whereas Appendix A.4 explicitly describes probabilistic completeness relying on assumptions such as (\epsilon)-greedy exploration and persistent exploration over sufficient time.

2. The reported 100% MAP is not strong evidence of perfect isomorphism prediction.

    Table 1 reports MAP (=1.000) on TU datasets with small graphs ((|V_Q|\le 15, |V_T|\le 20)), but MAP in such settings does not guarantee perfect isomorphism decision. The result appears more like a special case of “approximate retrieval + small candidate sets,” and it is unclear whether the method maintains such performance on larger graphs, weaker labels, or more challenging structural variations.

    More robust metrics (precision/recall, false positive/negative analysis, Top-(k) success rates) are needed.

3. Fairness of comparisons with exact solvers.

    Efficiency comparisons fix (|V(q)| = 8), which may be unfavorable to certain baselines such as GNN-PE.

    A more systematic study—scaling (|V(q)|) from 4 to 64 and analyzing varying graph density/label entropy—is necessary to confirm consistent advantages beyond a single configuration.

4. Extremely high training cost and unclear generalization ability.

    The pipeline requires 50k epochs of supervised pretraining plus 10k RL steps (with neighborhood resampling every 50 epochs), which is computationally expensive. The paper does not demonstrate out-of-distribution generalization—e.g., when structural distributions, label vocabularies, noise levels, or missing attributes differ between training and test.

**Questions:**

1. Completeness.

    Is the “provably complete” claim in the main text referring to deterministic completeness or probabilistic completeness?

    Do the assumptions in Appendix A.4 (such as (\epsilon)-greedy, persistent exploration, and Markov chain ergodicity) hold during test-time inference?

    If the deployed policy is purely greedy, does completeness still hold?

2. Evaluation metrics.

    In MAP evaluation, how do candidate-set construction, the number of negative samples, and graph sizes affect the reported MAP (=100%)?

    Please clarify whether the conclusion is sensitive to these design choices.

3. Training cost and generalization.

    What is the total training time and compute budget for the 50k+10k training pipeline?

    Has the model been tested on large graphs of unseen types without retraining?

4. Fairness of comparison settings.

    Does the performance degrade at larger query sizes?

    Please provide (|V(q)|)-scaling experiments and memory/time curves to evaluate robustness.

---

> ### Author Response · Authors · 2025-12-03
> **Response to Reviewer 8b3e---Part 1**
>
> We sincerely appreciate the reviewer's thoughtful analysis and constructive feedback on our work SIREN. We are particularly grateful for the recognition of our method's **well-structured and modular design**, **comprehensive empirical evaluation** and **discussion of completeness**, and we sincerely thank you for your support of our work. Below we address each concern systematically:
>
> ## Weakness 1 and Question 1
> ### Weakness 1 and Question 1.1
> > Ambiguity in the “completeness” claim.
> The main text repeatedly uses terms such as “provably complete” and “guarantees completeness,” whereas Appendix A.4 explicitly describes probabilistic completeness relying on assumptions such as $\epsilon$-greedy exploration and persistent exploration over sufficient time.
> Is the “provably complete” claim in the main text referring to deterministic completeness or probabilistic completeness?
>
> We thank the reviewer for raising the discussion on completeness. As clarified in Appendix A.4, our method is in fact probabilistically complete rather than deterministically complete. We have revised all references to "completeness" in the paper to **"probabilistic completeness"** to avoid overclaim.
> In our current implementation, we only incorporate masks of previously visited node pairs at each search state to prevent redundant additions. Under this design, the additional space complexity is O(|V_Q| + |V_T|), which does not guarantee deterministic completeness.
> Moreover, our approach can be easily modified to achieve deterministic completeness by maintaining, at each search node, a record of whether all feasible states have been fully explored. In that case, the space overhead would increase to O(|S| × (|V_Q| + |V_T|)), where |S| denotes the total number of visited states. This would lead to a linear growth in memory usage with the number of explored states. Considering scalability, we have chosen not to adopt this strategy. (In fact, our experiments indicate that such modification brings limited improvement in time efficiency.)
> A straightforward modification of this kind offers limited performance gains. Therefore, to achieve deterministic completeness, a more desirable approach would be to employ a hash-based method to store all fully explored states. However, this would also lead to a linear increase in memory usage as the search progresses. This represents a typical trade-off between space and time efficiency, which we intentionally avoid. To mitigate the issue of repeated exploration, we believe that employing a Q-value-based selection mechanism, such as Upper Confidence Bound (UCB), coupled with a specific incentive for encouraging exploration along less-visited paths, would be highly effective. This direction is one of our planned future improvements.
>
> ### Question 1.2
> > Do the assumptions in Appendix A.4 (such as $\epsilon$-greedy, persistent exploration, and Markov chain ergodicity) hold during test-time inference?
>
> The assumptions outlined in Appendix A.4 (such as the use of $\epsilon$-greedy exploration, persistent exploration, and Markov chain ergodicity) are satisfied during the training phase and continue to hold throughout test-time inference. These properties are thus guaranteed in both stages.
>
> ### Question 1.3
> > If the deployed policy is purely greedy, does completeness still hold?
>
> We thank the reviewer for raising the discussion on greedy policy. Adopting a purely greedy policy during inference can cause the search process to stall in a local optimum. Specifically, the policy may oscillate indefinitely by repeatedly selecting and then reversing the same high-Q-value action (e.g., adding and then removing a node pair) upon nearing a solution, without making actual progress. This occurs because the learned Q-values are not guaranteed to be globally optimal. In such a scenario, deterministic completeness cannot be satisfied.

---

> ### Author Response · Authors · 2025-12-03
> **Response to Reviewer 8b3e---Part 2**
>
> ## Weakness 2 and Question 2
> ### Question 2.1
> > W2&Q2: The reported 100% MAP is not strong evidence of perfect isomorphism prediction.
> Table 1 reports MAP (=1.000) on TU datasets with small graphs ((|V_Q|\le 15, |V_T|\le 20)), but MAP in such settings does not guarantee perfect isomorphism decision. The result appears more like a special case of “approximate retrieval + small candidate sets,” and it is unclear whether the method maintains such performance on larger graphs, weaker labels, or more challenging structural variations.
> More robust metrics (precision/recall, false positive/negative analysis, Top-(k) success rates) are needed.
> In MAP evaluation, how do candidate-set construction, the number of negative samples, and graph sizes affect the reported MAP (=100%)?
> Please clarify whether the conclusion is sensitive to these design choices.
>
> We thank the reviewer for the feedback regarding the quality of the solutions. As described in the Methods and Experiments sections of our paper, our approach guarantees perfect isomorphism decision by exhaustively exploring all solution sets for a given query graph $G_Q$ and target graph $G_T$. Therefore, for the evaluation task in Table 1, we can determine subgraph isomorphism exactly—by either stopping the search once a solution is found or by fully traversing the solution space. This procedure ensures that there are no false positives or false negatives.
> Factors such as candidate-set construction, the number of negative samples, and graph sizes do not affect the correctness of the decision; they only influence the computational time required to explore the solution space, particularly for more complex datasets.
> In accordance with your request, we have supplemented the experiments in Table 1 with more detailed results, including HITS@20, Mean Reciprocal Rank (MRR), and Precision@20, to strengthen the validity of our conclusions.
> Below, we elaborate on the experimental setup: for a given $G_Q$, a set of candidate $G_T$ graphs is evaluated to determine whether $G_Q$ is isomorphic to a subgraph of $G_T$. For certain distance-based approximate matching methods, we rank the results by ascending distance to provide the final judgment. The experimental results are presented as follows.
>
> **HITS@20**
> | Method | AIDS | MUTAG | PTC-FM | PTC-FR | PTC-MM | PTC-MR |
> | --- | --- | --- | --- | --- | --- | --- |
> | SimGNN | 0.115 | 0.121 | 0.221 | 0.152 | 0.142 | 0.103 |
> | GraphSim | 0.048 | 0.071 | 0.077 | 0.058 | 0.053 | 0.083 |
> | GEDGNN | 0.159 | 0.371 | 0.218 | 0.255 | 0.268 | 0.114 |
> | GOTSim | 0.137 | 0.168 | 0.238 | 0.143 | 0.210 | 0.224 |
> | ERIC | 0.275 | 0.362 | 0.387 | 0.370 | 0.375 | 0.393 |
> | GMN-Match | 0.392 | 0.465 | 0.455 | 0.477 | 0.422 | 0.485 |
> | NeuroMatch | 0.221 | 0.354 | 0.405 | 0.367 | 0.273 | 0.359 |
> | IsoNet | 0.482 | 0.568 | 0.601 | 0.553 | 0.571 | 0.588 |
> | SubMDSE-Late | 0.541 | 0.525 | 0.648 | 0.569 | 0.594 | 0.625 |
> | SubMDSF-Early | 0.677 | 0.726 | 0.785 | 0.730 | 0.716 | 0.752 |
> | SIREN | 1.000 | 1.000 | 1.000 | 1.000 | 1.000 | 1.000 |
>
> **Mean Reciprocal Rank (MRR)**
> | Method | AIDS | MUTAG | PTC-FM | PTC-FR | PTC-MM | PTC-MR |
> | --- | --- | --- | --- | --- | --- | --- |
> | SimGNN | 0.585 | 0.600 | 0.815 | 0.724 | 0.699 | 0.553 |
> | GraphSim | 0.345 | 0.401 | 0.492 | 0.357 | 0.402 | 0.480 |
> | GEDGNN | 0.725 | 0.870 | 0.791 | 0.846 | 0.856 | 0.576 |
> | GOTSim | 0.672 | 0.726 | 0.811 | 0.699 | 0.792 | 0.796 |
> | ERIC | 0.865 | 0.903 | 0.875 | 0.888 | 0.892 | 0.886 |
> | GMN-Match | 0.892 | 0.917 | 0.917 | 0.951 | 0.913 | 0.945 |
> | NeuroMatch | 0.793 | 0.862 | 0.896 | 0.885 | 0.854 | 0.887 |
> | IsoNet | 0.914 | 0.974 | 0.965 | 0.961 | 0.959 | 0.965 |
> | SubMDSE-Late | 0.962 | 0.946 | 1.000 | 0.988 | 1.000 | 1.000 |
> | SubMDSF-Early | 1.000 | 1.000 | 0.998 | 0.998 | 0.997 | 0.999 |
> | SIREN | 1.000 | 1.000 | 1.000 | 1.000 | 1.000 | 1.000 |
>
> **Precision@20**
> | Method | AIDS | MUTAG | PTC-FM | PTC-FR | PTC-MM | PTC-MR |
> | --- | --- | --- | --- | --- | --- | --- |
> | SimGNN | 0.409 | 0.423 | 0.614 | 0.481 | 0.467 | 0.391 |
> | GraphSim | 0.193 | 0.233 | 0.279 | 0.204 | 0.221 | 0.276 |
> | GEDGNN | 0.480 | 0.734 | 0.599 | 0.646 | 0.670 | 0.408 |
> | GOTSim | 0.450 | 0.527 | 0.616 | 0.472 | 0.597 | 0.604 |
> | ERIC | 0.681 | 0.721 | 0.767 | 0.708 | 0.711 | 0.762 |
> | GMN-Match | 0.765 | 0.801 | 0.799 | 0.828 | 0.800 | 0.826 |
> | NeuroMatch | 0.600 | 0.694 | 0.788 | 0.704 | 0.678 | 0.707 |
> | IsoNet | 0.797 | 0.876 | 0.886 | 0.862 | 0.870 | 0.877 |
> | SubMDSE-Late | 0.832 | 0.833 | 0.920 | 0.879 | 0.893 | 0.909 |
> | SubMDSF-Early | 0.916 | 0.941 | 0.931 | 0.919 | 0.911 | 0.929 |
> | SIREN | 1.000 | 1.000 | 1.000 | 1.000 | 1.000 | 1.000 |

---

> ### Author Response · Authors · 2025-12-03
> **Response to Reviewer 8b3e---Part 3**
>
> ## Weakness 3 and Question 4
> > W3&Q4: Fairness of comparisons with exact solvers.
> Efficiency comparisons fix (|V(q)| = 8), which may be unfavorable to certain baselines such as GNN-PE.
> A more systematic study—scaling (|V(q)|) from 4 to 64 and analyzing varying graph density/label entropy—is necessary to confirm consistent advantages beyond a single configuration.
> Does the performance degrade at larger query sizes?
> Please provide (|V(q)|)-scaling experiments and memory/time curves to evaluate robustness.
>
> We thank the reviewer for raising the point regarding fair comparison. We are committed to ensuring a fair evaluation of all existing methods, and within the limited time, we have made our best effort to supplement the detailed $|V_Q|$-scaling experiments as suggested.
> Regarding the reviewer’s interest in the GNN-PE method, we provide further discussion below. The GNN-PE approach consists of two main stages: an offline preprocessing phase and an online graph matching phase. The preprocessing stage involves data graph partitioning and the construction of index structures based on GNN and R-trees, while the online phase primarily performs parallel index traversal. However, due to its reliance on relatively complex indexing structures, GNN-PE is not well-suited for queries with large $|V_Q|$, as the space overhead of the R-tree index becomes substantial. In fact, in the original GNN-PE paper, the $|V_Q|$ values ranged only from 5 to 12, precisely because larger query graphs would lead to prohibitively high memory usage.
> In contrast, our method exhibits linear space complexity and does not incur additional memory overhead during the solution space exploration using DQN, which we consider a notable advantage. On the other hand, when $|V_Q|$ is very small and the query structure is simple (e.g., edgeless graphs), all existing methods—including ours—face the challenge of an excessively large solution space, making it time-consuming to enumerate all possible solutions. Therefore, we conducted experiments using query graphs of moderate structural complexity to ensure feasible evaluation.
> Below, we present the results (average time:s) of the detailed $|V_Q|$-scaling experiments.
>
> | V_Q | RI | VF | GraphQL | QuickSI | GNN-PE | VF3 | RM | VEQ | CaliG | DPIso | SIREN |
> | --- | --- | --- | --- | --- | --- | --- | --- | --- | --- | --- |  --- |
> | 4 | 2483.4 | 2795.5 | 14653.0 | 2030.7 | 87.3 | 828.5 | 101.2 | 111.8 | 609.6 | 131.1 | 50.7 |
> | 6 | 4966.3 | 3678.2 | 14412.6 | 2108.7 | 125.0 | 1306.7 | 203.2 | 120.4 | 689.8 | 173.3 | 79.2 |
> | 8 | 5415.2 | 4309.1 | 25414.6 | 3813.2 | 225.8 | 2143.7 | 285.6 | 180.1 | 1142.9 | 230.5 | 148.4 |
> | 12 | 11231.3 | 9013.4 | 48940.1 | 5314.4 | 265.3 | 3847.0 | 438.9 | 216.3 | 1822.8 | 255.5 | 301.6 |
> | 16 | 18212.3 | 19085.3 | 89025.2 | 15454.8 | 931.0 | 6791.3 | 1496.5 | 851.2 | 5897.7 | 1162.3 | 632.0 |
> | 64 | $>10^5$ | $>10^5$ | $>10^5$ | $>10^5$ | $>10^5$ | $>10^5$ | $>10^5$ | $>10^5$ | $>10^5$ | $>10^5$ | 65258.2 |
>
> In accordance with your request, we have also evaluated the inference cost of our method on large-scale graphs containing millions of nodes, along with the cost of our original training process. The accompanying figures present the GPU memory/time curves. A subset of the numerical results used to generate these curves is provided in the table below. The data clearly demonstrates that our method maintains low memory (Maximum cost ~ 10GB for ultra-large graph) and GPU overhead throughout, which is a notable advantage.
>
> **GPU Memory Usage (MB)**
> | Time | Test_Small(US Patents) | Test_Large(Synthetic) | Train |
> | --- | --- | --- | --- |
> | 0min | 573 | 763 | 437 |
> | 1min | 585 | 779 | 577 |
> | 2min | - | 803 | 601 |
> | 3min | - | 823 | 621 |
> | 4min | - | 1995 | 649 |
> | 5min | - | 1995 | 669 |
> | 6min | - | 1995 | 689 |
> | 7min | - | 2005 | 713 |
> | 8min | - | 2027 | 737 |
> | 9min | - | 9197 | 757 |
> | 10min | - | 9207 | 765 |
> | Maximum | 597 | 10244 | 2018 |

---

> ### Author Response · Authors · 2025-12-03
> **Response to Reviewer 8b3e---Part 4**
>
> ## Weakness 4 and Question 3
> > W4&Q3: Extremely high training cost and unclear generalization ability.
> The pipeline requires 50k epochs of supervised pretraining plus 10k RL steps (with neighborhood resampling every 50 epochs), which is computationally expensive. The paper does not demonstrate out-of-distribution generalization—e.g., when structural distributions, label vocabularies, noise levels, or missing attributes differ between training and test.
> What is the total training time and compute budget for the 50k+10k training pipeline?
> Has the model been tested on large graphs of unseen types without retraining?
>
> **About Training Cost**
> We thank the reviewer for raising the discussion on training cost. We provide the total training duration of our final model in the table below, including both the pretrained GNN phase and DQN training period. For the DQN component, we trained for 10000 iterations per dataset group. Across the 18 dataset groups utilized in our experiments (FR, FM, MM, MR, MUTAG, AIDS, FM, NCI, MOLT, Yeast, Human, HPRD, WordNet, DBLP, Youtube, US Patents, adversarial dense dataset, and synthetic dataset), we provide the total training duration and average time per iteration below. As specified in Section 4.2, all training was conducted utilizing 4 $\times$ NVIDIA A800 GPUs.
>
> **Training Time**
> | Period | Epochs/Iterations | est. Time per epoch/iter | Total Time |
> | --- | --- | --- | --- |
> | GNN Pretraining | 50000 Epochs | 1.33s | \~66433s (\~18.4h) |
> | DQN Training | 180000 Iterations | 0.23s | \~41709s (\~11.6h) |
> | Total |  |  | \~108142s (\~30.0h) |
>
> As demonstrated in the preceding table, SIREN incurs manageable computational overhead during its full training cycle, requiring only approximately 30 GPU-hours. This resource utilization leverages commercially accessible hardware: 4 $\times$ NVIDIA A800 GPUs.
>
> **About Adversarial Dataset**
> Our method is primarily designed for graphs without node labels, i.e., those containing only structural information, which represent a more challenging and information-sparse setting. Therefore, in simpler cases where node labels are available, even without additional training, the extra label information can assist our DQN in performing more effective pruning. As a result, factors such as label vocabularies, noise levels, or missing attributes do not adversely affect our approach. In fact, the presence of labels provides additional pruning cues that are inherently beneficial compared to the purely structural setting.
> Moreover, since our training set is highly diverse and covers a wide range of possible graph structures, no additional DQN training is generally required for standard datasets. We have further conducted comparisons on a new, strongly adversarial dataset comprising entirely unfamiliar and complex graph structures. The results indicate that while our method requires more exploration time in such extreme cases, it still maintains a noticeable efficiency advantage over existing methods.
>
> We believe that, in contrast to existing approaches which often entail high migration costs (e.g., GNN-PE with its expensive indexing overhead) or excessive memory consumption (e.g., VEQ), our method offers a highly favorable balance of training cost and overall efficiency, coupled with strong transferability.
>
> If the reviewers have **any remaining concerns** that have not been addressed, or if there are **any other suggestions** to further improve the paper, **please feel free to inform us at any time**.
>
> Thank you again for your thoughtful feedback and the valuable discussions.
>
> Best regards,
>
> The authors of Paper 17918

---

### Author Response · Authors · 2025-12-03
**A Concise Summary of Rebuttal**

Dear Area Chair,

Thank you for efficiently handing our submission and leading the rebuttal discussion. To assist your final assessment, we provide a concise summary of our rebuttal and the additional experiments conducted to address all concerns.

1. **Summary of Strengths**

Reviewers highlighted several core strengths of our work:

* **Novel and well-structured and methodological integration**: Well-structured and modular design integrates RL + GNN, which is novel, strong and practical (8b3e, qaH3, TGyc, 8kgs)
* **Theoretically grounded and elegant GNN model**: Theoretically grounded, clever and valuable introduction of the partial-order-aware GNN as a significant contribution (qaH3, 8b3e)
* **Comprehensive empirical evaluation with impressive results**: Comprehensive empirical evaluation across accuracy, efficiency, and scalability (qaH3), achieving 100% MAP and highly effective speedups (TGyc)
* **Discussion on probabilistic completeness**: Provides a theoretical discussion on probabilistic completeness which avoids portraying RL search as a black-box heuristic (8b3e, 8kgs)

2. **Summary of Rebuttal & Clarifications**

The reviewers' primary concerns involved theoretical discussion, training cost, comparison fairness, generalization and ablation. We addressed each via the following revisions and experiments:

* **Theoretical discussion on Completeness (8b3e, TGyc)**: We have clarified that our current method is probabilistically complete and provided a modification to achieve deterministic completeness, along with the reasons for not adopting this modification. We have supplemented the evaluation with additional metrics to demonstrate the completeness of our method (8b3e).
* **Training Cost(8b3e, qaH3)**: We have provided the memory consumption and training time of our method, demonstrating its favorable spatial scalability and acceptable training cost.
* **Comparison Fairness(8b3e, TGyc, 8kgs)**: We have added |V(q)|-scaling experiments to further demonstrate the effectiveness of our approach (8b3e), and have included additional experimental details to show that our comparisons with existing methods are fair (TGyc, 8kgs).
* **Generalization to Similar Problems or Labelled Graph(qaH3, 8kgs)**: We have supplemented the discussion with relevant ideas and preliminary results on generalizing our method to similar problems.
* **Ablation(TGyc, 8kgs)**: We have clarified that our optimal model relies on labels from heuristic methods, and have supplemented the discussion with ablation studies on both the DQN training phase and the GNN module.

3. **Summary of Revisions**

We updated the manuscript to incorporate these results and analyses:

* New Appendices:
* **Ablation Studies**: Added comprehensive ablation studies on the DQN training phase and the GNN module (A.14).
* **Additional Metrics**: Evaluation with additional metrics to demonstrate the completeness (A.12).
* **Additional Scaling Results**: Additional $|V_Q|$-scaling results (A.13).
* **Generalization to Similar Problems**: Discussions on extensions to related problems (A.15).
* Minor Corrections: "provable completeness" -> "probabilistic completeness".

We believe our detailed rebuttal has fully resolved the reviewers' concerns. We are devoted to delivering a revised manuscript that reflects these improvements and underscores the significant advance of our work. Thank you for your time and consideration.

Best regards,

The authors of Paper 17918

---

### Meta-Review · Area_Chair_ynbU · 2026-01-07

**Summary:**

Ambiguity in completeness claims: The paper initially states “provable/guaranteed completeness,” while the technical discussion clarifies that the method is only probabilistically complete under \epsilon-greedy and exploration assumptions, which differs from deterministic completeness.

Complex training pipeline and solver dependence: The approach requires large-scale pretraining and a multi-stage pipeline, and relies on VF3-generated demonstrations, which conflicts with claims of eliminating dependence on traditional solvers.

Evaluation design and comparison fairness: The interpretation of MAP=1.0 on small benchmark graphs was questioned along with the lack of more direct error-analysis metrics.
Fixed query sizes and differences in model capacity raised concerns about fair comparison with baselines.

Limited empirical evidence for generalization: Generalization to out-of-distribution graphs, noisy or labeled/heterogeneous graphs, and related tasks is mainly discussed conceptually with limited empirical validation.

Reproducibility concerns: Code is promised only upon acceptance

**Reviewer Concerns:**

Ambiguity in completeness claims: The rebuttal clarified that the method is probabilistically, not deterministically, complete and revised the wording accordingly, addressing the overstatement. The approach nevertheless remains probabilistic by design.

Complex training pipeline and solver dependence: The rebuttal provided concrete training cost details and acknowledged the reliance on VF3-based imitation learning and revised claims about eliminating solver dependence. The pipeline complexity and solver reliance remain inherent.

Evaluation design and comparison fairness: Additional metrics and results were added, which addressed concerns about MAP=1.0 and fixed query sizes. Some questions about full fairness across baselines and model capacity differences remain judgment-based.

Limited empirical evidence for generalization: The rebuttal added discussion and limited experiments on generalization, but broad empirical validation for OOD, noisy, or heterogeneous graphs remains incomplete.

Reproducibility concerns: The rebuttal reaffirmed code release upon acceptance, but independent verification is still limited at submission time.

**Reviewer Scores:**

Overall, the rebuttal addressed many clarification-level and technical concerns, which clarifies the completeness claims, adds concrete training cost information, and extends the evaluation with additional metrics and scaling experiments.
Reviewers who were initially positive would likely maintain their scores or gain slightly increased confidence, while borderline reviewers might shift modestly upward, as several major ambiguities were resolved.
Reviewers whose concerns were primarily judgment-based, such as the inherent complexity and solver dependence of the training pipeline, limited empirical evidence for broad generalization, and reproducibility constraints at submission time, would likely maintain their original scores, as these issues were acknowledged but not fully eliminated.

---

### Decision · Program_Chairs · 2026-01-26

Reject